# Explaining A Black-box By Using A Deep Variational Information Bottleneck Approach

## Abstract

Interpretable machine learning has gained much attention recently. Briefness and comprehensiveness are necessary in order to provide a large amount of information concisely when explaining a black-box decision system. However, existing interpretable machine learning methods fail to consider briefness and comprehensiveness simultaneously, leading to redundant explanations. We propose the variational information bottleneck for interpretation, VIBI, a system-agnostic interpretable method that provides a brief but comprehensive explanation. VIBI adopts an information theoretic principle, *information bottleneck principle*, as a criterion for finding such explanations. For each instance, VIBI selects key features that are maximally compressed about an input (briefness), and informative about a decision made by a black-box system on that input (comprehensive). We evaluate VIBI on three datasets and compare with state-of-the-art interpretable machine learning methods in terms of both interpretability and fidelity evaluated by human and quantitative metrics.

## 1 Introduction

Interpretability is crucial in building and deploying black-box decision systems such as deep learning models. Interpretation of a black-box system helps decide whether or not to follow its decisions, or understand the logic behind the system. In recent years, the extensive use of deep learning black-box systems has given rise to interpretable machine learning approaches (Lipton, 2016; Doshi-Velez & Kim, 2017), which aim to explain how black-box systems work or why they reach certain decisions. In order to provide sufficient information while avoiding redundancy when explaining a black-box decision, we need to consider both *briefness* and *comprehensiveness*. However, existing approaches lack in-depth consideration for and fail to find both brief but comprehensive explanation.

In order to obtain brief but comprehensive explanation, we adopt the *information bottleneck principle* (Tishby et al., 2000). This principle provides an appealing information theoretic perspective for learning supervised models by defining what we mean by a 'good' representation. The principle says that the optimal model transmits as much information as possible from its input to its output through a compressed representation called the information bottleneck. Then, the information bottleneck will maximally compress the mutual information (MI) with an input while preserving as much as possible MI with the output. Recently, it has been shown that the principle also applies to deep neural networks and each layer of a deep neural network can work as an information bottleneck (Tishby & Zaslavsky, 2015; Shwartz-Ziv & Tishby, 2017). Using this idea of information bottleneck principle, we define a brief but comprehensive explanation as maximally informative about the black-box decision while compressive about a given input.

In this paper, we introduce the variational information bottleneck for interpretation (VIBI), a system-agnostic information bottleneck model that provides a brief but comprehensive explanation for every single decision made by a black-box model. VIBI is composed of two parts: explainer and approximator, each of which is modeled by a deep neural network. The explainer returns a probability whether a chunk of features such as a word, phrase, sentence or a group of pixels will be selected as an explanation or not for each instance, and an approximator mimics behaviour of a black-box model. Using the information bottleneck principle, we learn an explainer that favors brief explanations while enforcing that the explanations alone suffice for accurate approximations to a black-box model.

## 1.1 CONTRIBUTION

Our main contribution is to provide a new framework that systematically defines and generates a 'good' (i.e. brief but comprehensive) explanation using the information bottleneck principle. Based on this principle, we develop VIBI that favors a brief but comprehensive explanation. In order to make the objective function of VIBI tractable, we derive a variational approximation to the objective.

The benefits of our method are as follows. 1) **System-agnostic**: VIBI can be applied to explain any black-box system. 2) **Post-hoc learning**: VIBI is learned in a post-hoc manner, hence there is no trade-off between task accuracy of a black-box system and interpretability of an explainer. 3) **Cognitive chunk**: Cognitive chunk is defined as a group of raw features whose identity is understandable to human. VIBI groups non-cognitive raw features such as a pixel and letter into a cognitive chunk (e.g. a group of pixels, a word, a phrase, a sentence) and selects each unit as an explanation. 4) **Separate explainer and approximator**: The explainer and approximator are designed for separated tasks so that we do not need to limit the approximator to have a simple structure, which may reduce the fidelity (the ability to imitate the behaviour of a black-box) of approximator.

## 2 RELATED WORK

Most prior interpretable machine learning methods have been focusing on local interpretation, which implies knowing the reasons why a black-box system makes a certain decision at a very local point of interest, and our work is also situated in this line. Existing methods can be categorized into system-specific and system-agnostic method. System-specific methods only explain certain black-box decision systems (e.g. using backpropagation algorithm, or having CNN structure), while system-agnostic methods explain any black-box decision systems.

**System-specific methods**. To measure a change of output with respect to changes of input is an intuitive way of obtaining feature attribution for the output. Using this idea, Zeiler & Fergus (2014), and Zintgraf et al. (2017) observe how outputs change when they make perturbations to each instance. Baehrens et al. (2010); Simonyan et al. (2013), and Smilkov et al. (2017) use computationally more efficient approaches; they measure change of output by propagating contributions through layers of a deep neural network towards an input than the perturbation. However, these approaches fail to detect the changes of output when the prediction function is flattened at the instance (Shrikumar et al., 2017), which leads to interpretations focusing on irrelevant features. In order to solve this problem, the layer-wise relevance propagation (Bach et al., 2015; Binder et al., 2016), DeepLIFT (Shrikumar et al., 2017), and Integrated Gradients (Sundararajan et al., 2017) compare the changes of output to its reference output. Ross et al. (2017) learn a more generalizable model as well as desirable explanations by constraining its explanations (i.e. input gradient) to match domain knowledge.

**System-agnostic methods**. The great advantage of system-agnostic interpretable machine learning methods over system-specific methods is that their usage is not restricted to a specific black-box system. One of the most well-known system-agnostic methods is LIME (Ribeiro et al., 2016). It explains the decision of an instance by locally approximating the black-box decision boundary around the instance with an inherently interpretable model such as sparse linear or decision trees. The approximator is learned by samples generated by perturbing a given instance. Lundberg & Lee (2017) proposed SHAP, a unified measure defined over the additive feature attribution scores in order to achieve local accuracy, missingness, and consistency. L2X (Chen et al., 2018) learns a stochastic map that selects instance-wise features that are most informative for black-box decisions. Unlike LIME and SHAP, which approximate local behaviors of a black-box system with a simple (linear) model, L2X does not put a limit on the structure of the approximator; hence it avoids losing fidelity of the approximator. As SHAP does, Dabkowski & Gal (2017); Fong & Vedaldi (2017), and Petsiuk et al. (2018) use sample perturbation but they rather learn or estimate desired perturbation masks than using perturbed samples to learn an approximator. Dabkowski & Gal (2017), and Fong & Vedaldi (2017) learn a smallest perturbation mask that alters black-box outputs as much as possible. Petsiuk et al. (2018) empirically estimate feature attribution as a sum of random masks weighted by class scores corresponding to masked inputs. Our method VIBI is similar with L2X in that both learn a stochastic explainer that returns a distribution over the subset of features given the input and performs instance-wise feature selection based on that. However, given the same number of explanations, our explainer favors both briefness and comprehensiveness while the L2X explainer favors comprehensiveness of the explanation and does not account for briefness.

## 3 METHOD

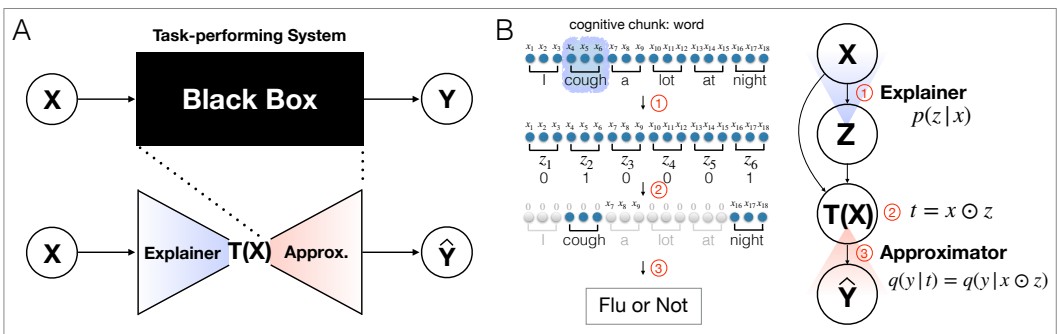

Figure 1: Illustration of VIBI. (A) VIBI is composed of two parts: the explainer and approximator. The explainer selects a group of $k$ key cognitive chunks given an instance while the approximator mimics the behaviour of the black-box system using the selected keys as the input. (B) We set each word as a cognitive chunk and $k = 2$. ① The explainer takes an input $\mathbf{x}$ and returns a stochastic k-hot random vector $\mathbf{z}$ which indicates whether each cognitive chunk will be selected as an explanation or not. ② $\mathbf{t}(\mathbf{x})$ provides instance-specific explanation. ③ The approximator takes $\mathbf{t}(\mathbf{x})$ as an input and approximates the black-box output.

### 3.1 PERSPECTIVE FROM INFORMATION BOTTLENECK PRINCIPLE

The information bottleneck principle (Tishby et al., 2000) provides an appealing information theoretic view for learning a supervised model by defining what we mean by a 'good' representation. The principle says that the optimal model transmits as much information as possible from the input $\mathbf{x}$ to the output $\mathbf{y}$ through a compressed representation $\mathbf{t}$ (called the information bottleneck). The representation $\mathbf{t}$ is stochastically defined and the optimal stochastic mapping $p(\mathbf{t}|\mathbf{x})$ is obtained by optimizing the following problem with a Markov chain assumption $\mathbf{y} \to \mathbf{x} \to \mathbf{t}$:

$$p(\mathbf{t}|\mathbf{x}) = \underset{p(\mathbf{t}|\mathbf{x}),p(\mathbf{y}|\mathbf{t}),p(\mathbf{t})}{\arg\max} \quad \mathrm{I}(\mathbf{t}, \mathbf{y}) - \beta\,\mathrm{I}(\mathbf{x}, \mathbf{t}) \tag{1}$$

where $\mathrm{I}(\cdot, \cdot)$ is the MI and $\beta$ is a Lagrange multiplier representing the trade-off between the compressiveness $-\mathrm{I}(\mathbf{x}, \mathbf{t})$ and informativeness $\mathrm{I}(\mathbf{t}, \mathbf{y})$ of the representation $\mathbf{t}$.

We adopt the information bottleneck principle as a criterion for finding brief but comprehensive explanations. Our aim is to learn an explainer generating explanations that are maximally informative about the black-box decision while compressive about a given input.

### 3.2 PROPOSED APPROACH

We introduce VIBI, a system-agnostic interpretation approach that provides brief but comprehensive explanations for decisions made by black-box decision system. In order to achieve this, we optimize the following information bottleneck objective.

$$p(\mathbf{z}|\mathbf{x}) = \underset{p(\mathbf{z}|\mathbf{x}),p(\mathbf{y}|\mathbf{t})}{\arg\max} \quad \mathrm{I}(\mathbf{t}, \mathbf{y}) - \beta\,\mathrm{I}(\mathbf{x}, \mathbf{t}) \tag{2}$$

where $\mathrm{I}(\mathbf{t}, \mathbf{y})$ represents the sufficiency of information retained for explaining the black-box output $\mathbf{y}$, $-\mathrm{I}(\mathbf{x}, \mathbf{t})$ represents the briefness of the explanation $\mathbf{t}$, and $\beta$ is a Lagrange multiplier representing a trade-off between the two. The primary difference between our information bottleneck objective (2) and the one in Tishby et al. (2000) is as follows: the latter aims to identify a stochastic map of the representation $\mathbf{t}$ that itself works as an information bottleneck, whereas our objective aims to identify a stochastic map of $\mathbf{z}$ performing instance-wise selection of cognitive chunks and define information bottleneck as a function of $\mathbf{z}$ and the input $\mathbf{x}$.

As illustrated in Figure 1A, VIBI is composed of two parts: the explainer and the approximator, each of which is modeled by a deep neural network. The explainer selects a group of $k$ key cognitive chunks given an instance while the approximator mimics the behaviour of the black-box system using

the selected keys as the input. $k$ controls the level of sparsity in $\mathbf{z}$. In detail, the explainer $p(\mathbf{z}|\mathbf{x}; \boldsymbol{\theta}_e)$ is a map from an input $\mathbf{x}$ to its attribution scores $p_j(\mathbf{x}) = p(\mathbf{z}_j|\mathbf{x})$ where $j$ is for the $j$-th cognitive chunk and $\mathbf{z}_j$ is a binary indicator whether the chunk will be selected or not. The attribution score indicates the probability that each cognitive chunk to be selected. In order to select top $k$ cognitive chunks as an explanation, a $k$-hot vector $\mathbf{z}$ is sampled from a categorical distribution with class probabilities $p_j(\mathbf{x}) = p(\mathbf{z}_j|\mathbf{x})$ and the $j$-th cognitive chunk is selected if $\mathbf{z}_j = 1$. More specifically, the explanation $\mathbf{t}$ is defined as follows:

$$\mathbf{t}_i = (\mathbf{x} \odot \mathbf{z})_i = \mathbf{x}_i \times \mathbf{z}_j,$$

where $j$ indicates a cognitive chunk, each of which corresponds to multiple row features $i$. The approximator is modeled by another deep neural network $p(\mathbf{y}|\mathbf{t}; \boldsymbol{\theta}_a)$, which mimics the black-box decision system. It takes $\mathbf{t}$ as an input and returns an output approximating the black-box output for the instance $\mathbf{x}$. $\boldsymbol{\theta}_a$ and $\boldsymbol{\theta}_e$ represent the weight parameters of neural networks. The explainer and approximator are trained jointly by minimizing a cost function that favors concise explanations while enforcing that the explanations alone suffice for accurate prediction.

To achieve compressiveness, in addition to encouraging small MI between explanations and inputs, we also encourage the number of selected cognitive chunks to be small, i.e., encouraging $\mathbf{z}$ to be sparse. Note that MI and sparsity are two complementary approaches for achieving compression. MI aims at reducing semantic redundancy in explanations. Sparsity cannot achieve such a goal. For example, consider a movie review where "great" occurs a lot and two explanations in judging the sentiment of the review: "great, great" and "great, thought-provoking". They have the same level of sparsity ($k = 2$), but the former has semantic redundancy. In this case, MI helps to choose a better explanation. The first explanation has a larger MI with the input document. The second explanation has smaller MI and hence is more brief and preferable.

### 3.2.1 THE VARIATIONAL BOUND

The current form of information bottleneck objective is intractable due to the MIs $I(\mathbf{t}, \mathbf{y})$ and $I(\mathbf{x}, \mathbf{t})$. We address this problem by using a variational approximation of our information bottleneck objective. In this section, we summarize the results and refer to Supplementary Material A for details.

**Variational bound for** $I(\mathbf{x}, \mathbf{t})$: We first show that $I(\mathbf{x}, \mathbf{t}) \leq I(\mathbf{x}, \mathbf{z}) + C$ where $C$ is constant and use the lower bound for $-I(\mathbf{x}, \mathbf{z}) - C$ as a lower bound for $-I(\mathbf{x}, \mathbf{t})$. As a result, we obtain:

$$I(\mathbf{x}, \mathbf{t}) \leq I(\mathbf{x}, \mathbf{z}) + C \leq \mathbb{E}_{(\mathbf{x}, \mathbf{z}) \sim p(\mathbf{x}, \mathbf{z})} \left[ \log \frac{p(\mathbf{z}|\mathbf{x})}{r(\mathbf{z})} \right] + C = \mathbb{E}_{\mathbf{x} \sim p(\mathbf{x})} D_{\mathrm{KL}}(p(\mathbf{z}|\mathbf{x}), r(\mathbf{z})) + C \quad (3)$$

Note that with proper choices of $r(\mathbf{z})$ and $p(\mathbf{z}|\mathbf{x})$, we can assume that the Kullback-Leibler divergence $D_{\mathrm{KL}}(p(\mathbf{z}|\mathbf{x}), r(\mathbf{z}))$ has an analytical form.

**Variational bound for** $I(\mathbf{t}, \mathbf{y})$: We obtain the lower bound for $I(\mathbf{t}, \mathbf{y})$ by using $q(\mathbf{y}|\mathbf{t})$ to approximate $p(\mathbf{y}|\mathbf{t})$, which works as an approximator to the black-box system. As a result, we obtain:

$$I(\mathbf{t}, \mathbf{y}) \geq \mathbb{E}_{(\mathbf{t}, \mathbf{y}) \sim p(\mathbf{t}, \mathbf{y})} \left[ \log q(\mathbf{y}|\mathbf{t}) \right] = \mathbb{E}_{\mathbf{x} \sim p(\mathbf{x})} \mathbb{E}_{\mathbf{y}|\mathbf{x} \sim p(\mathbf{y}|\mathbf{x})} \mathbb{E}_{\mathbf{t}|\mathbf{x} \sim p(\mathbf{t}|\mathbf{x})} \left[ \log q(\mathbf{y}|\mathbf{t}) \right] \quad (4)$$

where $p(\mathbf{t}|\mathbf{x}, \mathbf{y}) = p(\mathbf{t}|\mathbf{x})$ by the Markov chain assumption $\mathbf{y} \leftrightarrow \mathbf{x} \leftrightarrow \mathbf{t}$.

Combining Equations (3) and (4), we obtain the following variational bound:

$$\begin{aligned} I(\mathbf{t}, \mathbf{y}) &- \beta\, I(\mathbf{x}, \mathbf{t}) \\ &\geq \mathbb{E}_{\mathbf{x} \sim p(\mathbf{x})} \mathbb{E}_{\mathbf{y}|\mathbf{x} \sim p(\mathbf{y}|\mathbf{x})} \mathbb{E}_{\mathbf{t}|\mathbf{x} \sim p(\mathbf{t}|\mathbf{x})} \left[ \log q(\mathbf{y}|\mathbf{t}) \right] - \beta\, \mathbb{E}_{\mathbf{x} \sim p(\mathbf{x})} D_{\mathrm{KL}}(p(\mathbf{z}|\mathbf{x}), r(\mathbf{z})) + C^*. \quad (5) \end{aligned}$$

where $C^* = -C\beta$ can be ignored since it is independent of the optimization procedure. We use the empirical data distribution to approximate $p(\mathbf{x}, \mathbf{y}) = p(\mathbf{x})p(\mathbf{y}|\mathbf{x})$ and $p(\mathbf{x})$.

### 3.2.2 CONTINUOUS RELAXATION AND REPARAMETERIZATION

Current form of the bound (5) is still intractable because we need to sum over the $\binom{d}{k}$ combinations of feature subsets. This is because we sample top $k$ out of $d$ cognitive chunks where each chunk is assumed drawn from a categorical distribution with class probabilities $p_j(\mathbf{x}) = p(\mathbf{z}_j|\mathbf{x})$. In order to avoid this, we use the generalized Gumbel-softmax trick (Jang et al., 2017; Chen et al., 2018). This is

a well-known technique that are used to approximate a non-differentiable categorical subset sampling with differentiable Gumbel-softmax samples. The steps are as follows.

First, we independently sample a cognitive chunk for $k$ times. For each time, a random perturbation $\mathrm{e}_j$ is added to the log probability of each cognitive chunk $\log p_j(\mathbf{x})$. From this, Concrete random vector $\mathbf{c} = (\mathrm{c}_1, \cdots, \mathrm{c}_d)$ working as a continuous, differentiable approximation to argmax is defined:

$$\mathrm{g}_j = -\log\left(-\log \mathrm{e}_j\right) \quad \text{where } \mathrm{e}_j \sim U(0,1)$$

$$\mathrm{c}_j = \frac{\exp\left(\left(\mathrm{g}_j + \log p_j(\mathbf{x})\right)/\tau\right)}{\sum_{j=1}^{d} \exp\left(\left(\mathrm{g}_j + \log p_j(\mathbf{x})\right)/\tau\right)},$$

where $\tau$ is a tuning parameter for the temperature of Gumbel-Softmax distribution. Next, we define a continuous-relaxed random vector $\mathbf{z}^* = [\mathrm{z}_1^*, \cdots, \mathrm{z}_d^*]^\top$ as the element-wise maximum of the independently sampled Concrete vectors $\mathbf{c}^{(l)}$ where $l = 1, \cdots, k$:

$$\mathrm{z}_j^* = \max_l \mathrm{c}_j^{(l)} \quad \text{for } l = 1, \cdots, k$$

With this sampling scheme, we approximate the $k$-hot random vector and have the continuous approximation to the variational bound (5). This trick allows using standard backpropagation to compute the gradients of the parameters via reparameterization.

By putting everything together, we obtain:

$$\frac{1}{NL} \sum_{n}^{N} \sum_{l}^{L} \left[ \log q(\boldsymbol{y}_{(n)}|\mathbf{x}_{(n)} \odot f(\mathbf{e}_{(n)}^{(l)}, \mathbf{x}_{(n)})) - \beta\, D_{\mathrm{KL}}(p(\mathbf{z}_{(n)}^*|\mathbf{x}_{(n)}), r(\mathbf{z}_{(n)}^*)) \right]$$

where $N$ is the number of samples, $n$ indicate the $n$-th sample, $f(\mathbf{e}_{(n)}^{(l)}, \mathbf{x}_{(n)}) = \mathbf{z}_{(n)}^*$, $q(\boldsymbol{y}_{(n)}|\mathbf{x}_{(n)} \odot \mathbf{z}_{(n)}^*)$ is the approximator to the black-box system and $-D_{\mathrm{KL}}(p(\mathbf{z}^*|\mathbf{x}_{(n)}), r(\mathbf{z}^*))$ represents the compactness of the explanation. Once we learn the model, the attribution score $p_j(\mathbf{x})$ for each cognitive chunk is used to select top $k$ key cognitive chunks that are maximally compressive about the input $\mathbf{x}$ and informative about the black-box decision $\mathbf{y}$ on that input.

## 4 EXPERIMENTS

We evaluated VIBI on three datasets and compared with state-of-the-art interpretable machine learning methods. The evaluation is performed from two perspectives: *interpretability* and *fidelity*. The interpretability indicates the ability to explain a black-box model with human understandable terms. The fidelity implies how accurately our approximator approximates the black-box model. Based on these criteria, we compared VIBI with three state-of-the-art system-agnostic methods (LIME (Ribeiro et al., 2016), SHAP (Lundberg & Lee, 2017) and L2X (Chen et al., 2018)), and a commonly used model-specific method called Saliency Map (Simonyan et al., 2013). For Saliency Map, we used the smooth gradient technique (Smilkov et al., 2017) to get visually sharp gradient-based sensitivity maps over the basic gradient saliency map. See Supplementary Material B for further experimental details.

We examined how VIBI performs across different experimental settings varying the number of selected chunks $k$ (amount or number of explanation), size of chunk (unit of explanation), and trade-off parameter $\beta$ (trade-off between the compressiveness of explanation and information preserved about the output). The settings of hyperparameter tuning include (bold indicate the choice for our final model): the temperature for Gumbel-softmax approximation $\tau - \{0.1, 0.2, 0.5, \mathbf{0.7}, 1\}$, learning rate $- 5 \times 10^{-3}, 10^{-3}, 5 \times 10^{-4}, \mathbf{10^{-4}}, 5 \times 10^{-5}\}$ and $\beta - \{0, 0.001, \mathbf{0.01}, 0.1, 1, 10, 100\}$. We use Adam algorithm (Kingma & Ba, 2014) with batch size 100 for MNIST and 50 for IMDB, the coefficients used for computing running averages of gradient and its square $(\beta_1, \beta_2) = (0.5, 0.999)$, and $\epsilon = 10^{-8}$. We tuned the hyperparameters via grid search and picked up the hyperparameters that yield the best fidelity score on the validation set. The code is publicly available on GitHub `https://github.com/XXX`.[1]

Negative Sentiment if any negative words are found

I do NOT understand why anyone would **waste** their time or money on utter trash like this … Don't get me wrong -- I LOVE a good Western -- Notice I said "GOOD" -- this is **just** trash. The acting is **horrible** -- Val **Kilmer** must know someone or **owed** a favor or something for them just to use his face and name in this ridiculous piece of crap...
*True: Negative / B-Box: Negative*

Positive Sentiment if any positive words are found

I watched this movie when it was released and being really young and not too much into cinema it was one of the **most fascinating** cinematic experiences I ever had and it really left a mark inside me. At first I didn't quite **understand** the story and probably failed to … He plays so well the man that falls in love slowly but so deeply with Katherine **Clifton**, opens up his heart and dives into this prohibited **affair**….
*True: Positive / B-Box: Positive*

Negative sentiment but predicted as Positive because several positive words are found

The reality of the **mafia** environment is absolutely dog-eat-**dog** where a gangster will be killed for showing any sign of weakness because they become a liability. I've got no problem with the human side of gansters' being portrayed but Bugsy steers too far in the direction of soft, comical, men. The film is **enjoyable** but is **only** light entertainment and not a biopic of a man who, though **exciting**, was extremely dangerous and fearsome. The acting's all good and the direction very solid. The locations and era are very well represented and the themes very interesting….
*True: Negative / B-Box: Positive*

Figure 2: The movie reviews and explanations provided by VIBI were randomly selected from the validation set. The selected words are colored red. Each word is used as a cognitive chunk and $k = 5$ words are provided for each review.

## 4.1 LSTM MOVIE SENTIMENT PREDICTION MODEL USING IMDB

The IMDB (Maas et al., 2011) is a large text dataset containing movie reviews labeled by sentiment (positive/negative). We grouped the reviews into training, validation, and test sets, which have 25,000, 12,500, and 12,500 reviews respectively. Then, we trained a hierarchical LSTM for sentiment prediction, which has two LSTM layers where each layer encodes words and sentences respectively. It achieved 87% of test accuracy. In order to explain this LSTM black-box model, we applied VIBI. We parameterized the explainer using a bidirectional LSTM and approximator using a 2D CNN. For the details of the black-box model and VIBI architectures, see Supplementary Material B.1.

VIBI explains why the LSTM predicts each movie review to be positive/negative and provides instance-wise key words that are the most important attributes to the sentiment prediction. As seen in the top-right and top-left of Figure 2, VIBI shows that the positive (or negative) words pass through the bottleneck and make a correct prediction. The bottom of Figure 2 shows that the LSTM sentiment prediction model makes a wrong prediction for a negative review because the review includes several positive words such as 'enjoyable' and 'exciting'.

## 4.2 CNN DIGIT RECOGNITION MODEL USING MNIST

Same digit with different angles.   Different digits, upside down   Difference between 7 and 1   Same digit similar features

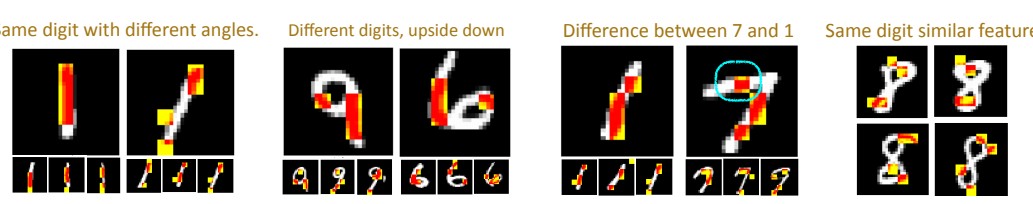

Figure 3: The hand-written digits and explanations provided by VIBI were randomly selected from the validation set. The selected patches are colored red if the pixel is activated (i.e. white) and yellow otherwise (i.e. black). A patch composed of $4 \times 4$ pixels is used as a cognitive chunk and $k = 4$ patches are identified for each image.

The MNIST (LeCun et al., 1998) is a large dataset contains $28 \times 28$ sized images of handwritten digits (0 to 9). We grouped the images into training, validation, and test sets, which have 50,000, 10,000, and 10,000 images respectively, and trained a simple 2D CNN for the digit recognition, which achieved 97% of test accuracy. In order to explain this CNN black-box model, we applied VIBI. We parameterized each the explainer and approximator using a 2D CNN. For the details of the black-box model and VIBI architectures, see Supplementary Material B.2.

VIBI explains how the CNN characterizes a digit and recognizes differences between digits. The first two examples in Figure 3 show that the CNN recognizes digits using both shapes and angles. In the first example, the CNN characterizes '1's by straightly aligned patches along with the activated

---

[1]Blinded due to the double-blind reviewing.

regions although '1's in the left and right panels are written at different angles. Contrary to the first example, the second example shows that the CNN recognizes the difference between '9' and '6' by their differences in angles. The last two examples in Figure 3 show that the CNN catches a difference of '7's from '1's by patches located on the activated horizontal line on '7' (see the cyan circle) and recognizes '8's by two patches on the top of the digits and another two patches at the bottom circle. More qualitative examples for VIBI and the baselines are shown in Supplementary Figure 6.

The briefness of explanations also depends on the sparsity $k$. Supplementary Figure 8 shows how our method works under different sparsity. When we increase $k$, VIBI tends to select patches that are the same with or nearby previously selected patches and additionally select patches that catch new characteristics of digits.

### 4.3 TCR TO EPITOPE BINDING PREDICTION MODEL USING VDJDB AND IEDB

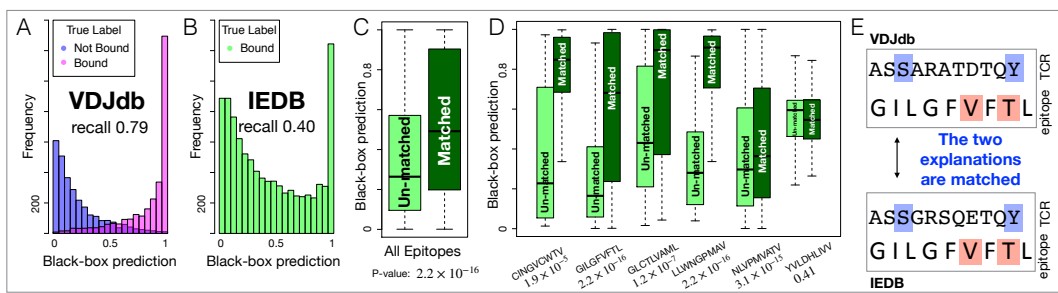

Figure 4: Black-box prediction scores of (A) VDJdb and (B) IEDB. (C) Black-box prediction scores between the matched and unmatched instances from IEDB and (D) those by six epitope sequences. (E) An example of matched explanation (The VIBI selected amino acids are shaded).

We next illustrate how VIBI can be used to get insights from a model and ensure the safety of a model in a real world application. Identifying which T-cell receptor (TCR) will bind to a specific epitope (i.e. cancer induced peptide molecules presented by the major histocompatibility complex to T-cells) is important for screening T-cells or genetically engineering T-cells that are effective in recognizing and destroying tumor cells. Therefore, there has been efforts in developing computational methods to predict binding affinity of given TCR-epitope pairs (Jurtz et al., 2018; Jokinen et al., 2019). These approaches rely on known interacting TCR-epitope pairs available from VDJdb (Shugay et al., 2017) and IEDB (Vita et al., 2014), which are the largest databases of several thousand entries. However, the number of unique TCRs harbored in a single individual is estimated to be $10^{10}$ (Lythe et al., 2016) and a theoretical number of epitopes of length $l$ is $20^l$, which are much larger than the number of known interacting TCR-epitope pairs.

One of the main concerns is whether a black-box model trained on such limited dataset can accurately predict TCR-epitope bindings of out-of-samples. This concern becomes pressing in a TCR-epitope binding prediction model trained on VDJdb (For details of the data, black-box model architecture, and parameter tuning, see Supplementary Material B.3). The model accurately predicted the (in-sample) bindings from VDJdb (recall 0.79, Figure 4A). However, it achieved poor prediction performance when it is used to predict the (out-of-sample) bindings from another dataset, IEDB (recall 0.40, Figure 4B). In an attempt to address this problem, we applied VIBI and determined whether or not to accept a decision made by the black-box model based on VIBI's explanation. As illustrated in Figure 4E, VIBI provided *matched explanations*—the identical amino acids in same positions (S and Y in this example) are highlighted in different TCR sequences when they are bound to the same epitope (GILGFVFTL in this example). Moreover, we found that if two TCR sequences binding to the same epitope, each from IEDB and VDJdb, are assigned with matched explanations by VIBI, then it significantly better predicts the binding than the others with no matching TCRs (Figure 4C-D, p-values are shown). Therefore, if a TCR sequence from IEDB has a matched explanation to a TCR from VDJdb, then we safely follow the positive decision made by the black-box model.

## 4.4 INTERPRETABILITY EVALUATED BY HUMANS

We evaluated interpretability of the methods on the LSTM movie sentiment prediction model and the CNN digit recognition model. For the movie sentiment prediction model, we provided instances that the black-box model had correctly predicted and asked humans to infer the output of the primary sentiment of the movie review (Positive/Negative/Neutral) given five key words selected by each method. Each method was evaluated by the humans on MTurk[2] who are awarded the Masters Qualification, high-performance workers who have demonstrated excellence across a wide range of tasks). We randomly selected and evaluated 200 instances for VIBI and 100 instances for the others. Five workers were assigned per instance. For the digit recognition model, we asked humans to directly score the explanation on a 0–5 scale. Each method was evaluated by 16 graduate students at XXX-University[3] who have taken at least one graduate-level machine learning class. For each method, 100 instances were randomly selected and evaluated. Four cognitive chunks with the size $4 \times 4$ were provided as an explanation for each instance ($\beta = 0.1$ for VIBI). On average, 4.26 students were assigned per instance. Further details regarding the experiments can be found in Supplementary Material C.

Table 1: Evaluation of interpretability

|  | Saliency | LIME | L2X | VIBI (Ours) |
|---|---|---|---|---|
| IMDB | 34.2% | 33.8% | 35.6% | **44.7%** |
| MNIST | 3.448 | 1.369 | 1.936 | **3.526** |

For IMDB, the percentage indicates how well the MTurk worker's answers match the black-box output. For MNIST, the score indicates how well the highlighted chunks catch key characteristics of handwritten digits. The average scores over all samples is shown on a 0 to 5 scale. See the survey example and detailed result in Supplementary Material Tables 3 and 4 for the detailed result.

As shown by the Table 1, VIBI better explains the black-box models. When explaining the movie sentiment prediction model, humans better inferred the (correctly predicted) black-box output given the five keywords when they were provided by VIBI. Therefore, it better captures the most contributing key words to the LSTM decision and better explains why the LSTM predicted each movie review by providing five key words. For explaining the digit recognition model, VIBI also highlighted the most concise chunks for explaining key characteristics of handwritten digit. Thus, it better explains how the CNN model recognized each the handwritten digit.

## 4.5 FIDELITY

Table 2: Evaluation of approximator and rationale fidelity

|  | chunk size | k | Approximator Fidelity | | | | | Rationale Fidelity | |
|---|---|---|---|---|---|---|---|---|---|
|  |  |  | Saliency | LIME | SHAP | L2X | VIBI (Ours) | L2X | VIBI (Ours) |
| IMDB | sentence | 1 | 38.7 ± 0.9 | 72.7 ± 0.8 | 49.5 ± 1.0 | 87.6 ± 0.6 | **87.7 ± 0.6** | 72.7 ± 0.8 | **73.1 ± 0.8** |
|  | word | 5 | 41.9 ± 0.9 | 75.6 ± 0.8 | 50.1 ± 1.0 | 73.8 ± 0.8 | **74.4 ± 0.8** | 63.8 ± 0.8 | **65.7 ± 0.8** |
|  | 5 words | 1 | 42.4 ± 0.9 | 29.0 ± 0.8 | 49.7 ± 1.0 | 75.9 ± 0.7 | **76.4 ± 0.7** | 60.1 ± 0.9 | **63.2 ± 0.8** |
|  | 5 words | 3 | 41.4 ± 0.9 | 67.9 ± 0.8 | 49.1 ± 1.0 | 83.3 ± 0.7 | **83.5 ± 0.7** | 69.4 ± 0.8 | 66.0 ± 0.8 |
| MNIST | 2 × 2 | 16 | 91.2 ± 0.6 | 77.0 ± 0.8 | 94.2 ± 0.5 | 93.4 ± 0.5 | **94.8 ± 0.4** | 73.5 ± 0.9 | **77.1 ± 0.8** |
|  | 2 × 2 | 24 | 93.8 ± 0.5 | 80.7 ± 0.8 | 95.4 ± 0.4 | 95.1 ± 0.4 | **95.3 ± 0.4** | 77.6 ± 0.8 | **85.6 ± 0.7** |
|  | 2 × 2 | 40 | 95.7 ± 0.4 | 85.9 ± 0.7 | 95.4 ± 0.4 | 96.7 ± 0.4 | **96.2 ± 0.4** | 81.1 ± 0.8 | **91.5 ± 0.5** |
|  | 4 × 4 | 4 | 86.3 ± 0.7 | 60.9 ± 1.0 | 94.8 ± 0.4 | 95.3 ± 0.4 | **94.8 ± 0.4** | 65.0 ± 0.9 | **77.5 ± 0.8** |
|  | 4 × 4 | 6 | 90.6 ± 0.6 | 63.7 ± 0.9 | 93.6 ± 0.5 | 95.7 ± 0.4 | **95.6 ± 0.4** | 51.1 ± 1.0 | **70.1 ± 0.9** |
|  | 4 × 4 | 10 | 94.9 ± 0.4 | 70.5 ± 0.9 | 95.1 ± 0.4 | 96.5 ± 0.4 | **96.7 ± 0.4** | 83.5 ± 0.7 | **93.3 ± 0.5** |

Note that the black-box models achieved 87% accuracy for IMDB and 97% accuracy for MNIST. $\beta = 0.1$ for VIBI. Accuracy and 0.95 confidence interval is shown. We performed three runs for each method and reported the best results. See more evaluations using F1-score and further results from different parameter settings in Supplementary Material Table 6 and 8 for approximator fidelity and Table 5 and 7 for rationale fidelity.

We assessed fidelity of the methods in approximating the black-box output. First, we compared the ability of the approximators to imitate behaviour of the black-box, denoted as *Approximator fidelity*.

---

[2]Amazon Mechanical Turk, https://www.mturk.com/

[3]Blinded due to the double-blinding reviewing

(See Supplementary Material B.4 for details about how each approximator fidelity is evaluated.) As shown in Table 2, VIBI has a better approximator fidelity than Saliency, LIME and SHAP in most cases. VIBI and L2X showed similar levels of approximator fidelity, so we further compared them based on *Rationale fidelity*. The difference between approximator and rationale fidelity is as follows. Approximator fidelity is quantified by prediction performance of the approximators that takes $\mathbf{t}^*$, the continuous relaxation of $\mathbf{t}$, as an input and the black-box output as a targeted label; rationale fidelity is quantified by using $\mathbf{t}$ instead of $\mathbf{t}^*$. Note that $\mathbf{t}$ only takes the top $k$ chunks and sets the others to be zero, while $\mathbf{t}^*$ sets the others to be small, non-zero values. Therefore, rationale fidelity allows to evaluate how much information purely flows through the explanations, not through a narrow crack made during the continuous relaxation procedure. As shown in Table 2, VIBI has a better rationale fidelity than L2X in most cases. Note that L2X can be viewed as a special case of VIBI without the compressiveness term, i.e., $\beta = 0$. The rationale fidelity empirically demonstrates that the compressiveness term can help the information to flow purely through the explanations.

## 5    CONCLUSION

We employ the information bottleneck principle as a criterion for learning 'good' explanations. Instance-wisely selected cognitive chunks work as an information bottleneck, hence, provide concise but comprehensive explanations for each decision made by a black-box system. The information bottleneck framework provides a theoretical background that the bottleneck captures a minimal sufficient statistic, i.e. the most compressed representation that captures all the possible (i.e. sufficient) amount of information about output. For finite $\beta$, the bottleneck approximates such a minimal sufficient statistic.

However, the way this information is represented may have a substantial effect on interpretability. VIBI helps to address this issue to some extent by always returning a certain form of output (i.e., a $k$-hot vector $\mathbf{z}$ assigned to each chunk) and having a certain form of the information bottleneck layer (i.e., a masked input) so that it makes sure that the explanations are easily understandable to humans. In practice, such a chunking strategy leads to a deviation from the strict theory that a 'good' explanation is the most compressed one but helps to achieve better interpretability in practice.

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

## A  VARIATIONAL INFERENCE TO CONSTRUCT A LOWER BOUND ON THE IB OBJECTIVE

We illustrate the variational approximation of the following information bottleneck objective:

$$p(\mathbf{t}|\mathbf{x}) = \underset{p(\mathbf{t}|\mathbf{x}),p(\mathbf{y}|\mathbf{t}),p(\mathbf{t})}{\arg\max} \quad \mathrm{I}(\mathbf{t},\mathbf{y}) - \beta\,\mathrm{I}(\mathbf{x},\mathbf{t})$$

Let us start with the Markov chain assumption $\mathbf{y} \to \mathbf{x} \to \mathbf{t}$, which also implies $\mathbf{y} \leftarrow \mathbf{x} \leftarrow \mathbf{t}$. Therefore, the condition is written as $\mathbf{y} \leftrightarrow \mathbf{x} \leftrightarrow \mathbf{t}$ (Cover & Thomas, 2012). Using this condition, we factorize the joint distribution $p(\mathbf{y},\mathbf{x},\mathbf{t})$:

$$p(\mathbf{y},\mathbf{x},\mathbf{t}) = p(\mathbf{x})p(\mathbf{t}|\mathbf{x})p(\mathbf{y}|\mathbf{t},\mathbf{x}) = p(\mathbf{x})p(\mathbf{t}|\mathbf{x})p(\mathbf{y}|\mathbf{t})$$

where $p(\mathbf{y}|\mathbf{t},\mathbf{x}) = p(\mathbf{y}|\mathbf{t})$ by the Markov chain.

Now, we will examine each of the expressions in the information bottleneck objective $\mathrm{I}(\mathbf{t},\mathbf{y}) - \beta\,\mathrm{I}(\mathbf{x},\mathbf{t})$ in turn.

### (I) VARIATIONAL LOWER BOUND FOR $\mathrm{I}(\mathbf{x},\mathbf{t})$

We first show that $\mathrm{I}(\mathbf{x},\mathbf{t}) \le \mathrm{I}(\mathbf{x},\mathbf{z}) + C$ where $C$ is a constant and then use the lower bound for $-\mathrm{I}(\mathbf{x},\mathbf{z}) - C$ as a lower bound for $-\mathrm{I}(\mathbf{x},\mathbf{t})$.

First, we prove $\mathrm{I}(\mathbf{x},\mathbf{t}) \le \mathrm{I}(\mathbf{x},\mathbf{z}) + C$. From the Markov Chain $\mathbf{x} \to (\mathbf{x},\mathbf{z}) \to \mathbf{t}$, we have $\mathrm{I}(\mathbf{x},\mathbf{t}) \le \mathrm{I}(\mathbf{x},(\mathbf{x},\mathbf{z}))$. According to the chain rule for mutual information, $\mathrm{I}(\mathbf{x},(\mathbf{x},\mathbf{z})) = \mathrm{I}(\mathbf{x},\mathbf{z}) + \mathrm{I}(\mathbf{x},\mathbf{x}|\mathbf{z})$, where $\mathrm{I}(\mathbf{x},\mathbf{x}|\mathbf{z}) = \mathrm{H}(\mathbf{x}|\mathbf{z}) + \mathrm{H}(\mathbf{x}|\mathbf{z}) - \mathrm{H}(\mathbf{x},\mathbf{x}|\mathbf{z})$. Further, $\mathrm{H}(\mathbf{x}|\mathbf{z}) \le \mathrm{H}(\mathbf{x})$. Putting these pieces together, we have

$$\mathrm{I}(\mathbf{x},\mathbf{t}) \le \mathrm{I}(\mathbf{x},\mathbf{z}) + \mathrm{H}(\mathbf{x}) \tag{6}$$

where entropy $\mathrm{H}(\mathbf{x})$ of input is a constant. For simplicity, we denote it as $C$.

We then approximate $p(\mathbf{z})$ using $r(\mathbf{z})$. From the fact that Kullback Leibler divergence is always positive, we have

$$
\begin{aligned}
\mathbb{E}_{(\mathbf{x},\mathbf{z})\sim p(\mathbf{x},\mathbf{z})}\left[\log p(\mathbf{z})\right] &= \mathbb{E}_{\mathbf{z}\sim p(\mathbf{z})}\left[\log p(\mathbf{z})\right] \\
&\ge \mathbb{E}_{\mathbf{z}\sim p(\mathbf{z})}\left[\log r(\mathbf{z})\right] \\
&= \mathbb{E}_{(\mathbf{x},\mathbf{z})\sim p(\mathbf{x},\mathbf{z})}\left[\log r(\mathbf{z})\right].
\end{aligned}
\tag{7}
$$

From (6) and (7), we have

$$
\begin{aligned}
\mathrm{I}(\mathbf{x}, \mathbf{t}) &\leq \mathrm{I}(\mathbf{x}, \mathbf{z}) + C \\
&= \mathbb{E}_{(\mathbf{x}, \mathbf{z}) \sim p(\mathbf{x}, \mathbf{z})} \left[ \log \frac{p(\mathbf{z}|\mathbf{x})}{p(\mathbf{z})} \right] + C \\
&\leq \mathbb{E}_{(\mathbf{x}, \mathbf{z}) \sim p(\mathbf{x}, \mathbf{z})} \left[ \log \frac{p(\mathbf{z}|\mathbf{x})}{r(\mathbf{z})} \right] + C \\
&= \mathbb{E}_{\mathbf{x} \sim p(\mathbf{x})} D_{\mathrm{KL}}(p(\mathbf{z}|\mathbf{x}), r(\mathbf{z})) + C.
\end{aligned}
$$

(II) VARIATIONAL LOWER BOUND FOR $\mathrm{I}(\mathbf{t}, \mathbf{y})$

Starting with $\mathrm{I}(\mathbf{t}, \mathbf{y})$, we have

$$
\begin{aligned}
\mathrm{I}(\mathbf{t}, \mathbf{y}) &= \mathbb{E}_{(\mathbf{t}, \mathbf{y}) \sim p(\mathbf{t}, \mathbf{y})} \left[ \log \frac{p(\mathbf{y}, \mathbf{t})}{p(\mathbf{y}) p(\mathbf{t})} \right] \\
&= \mathbb{E}_{(\mathbf{t}, \mathbf{y}) \sim p(\mathbf{t}, \mathbf{y})} \left[ \log \frac{p(\mathbf{y}|\mathbf{t})}{p(\mathbf{y})} \right] \\
&= \mathbb{E}_{(\mathbf{t}, \mathbf{y}) \sim p(\mathbf{t}, \mathbf{y})} \left[ \log p(\mathbf{y}|\mathbf{t}) \right] - \mathbb{E}_{\mathbf{y} \sim p(\mathbf{y})} \left[ \log p(\mathbf{y}) \right] \\
&= \mathbb{E}_{(\mathbf{t}, \mathbf{y}) \sim p(\mathbf{t}, \mathbf{y})} \left[ \log p(\mathbf{y}|\mathbf{t}) \right] + \mathrm{Const}.
\end{aligned}
$$

Note that $\mathrm{H}(\mathbf{y}) = -\mathbb{E}_{\mathbf{y} \sim p(\mathbf{y})} \left[ \log p(\mathbf{y}) \right]$ is independent of the optimization procedure, hence can be ignored. Now, we use $q(\mathbf{y}|\mathbf{t})$ to approximate $p(\mathbf{y}|\mathbf{t})$ which works as an approximator to the black-box system. Using the fact that Kullback Leibler divergence is always positive, we have

$$
\mathbb{E}_{\mathbf{y}|\mathbf{t} \sim p(\mathbf{y}|\mathbf{t})} \left[ \log p(\mathbf{y}|\mathbf{t}) \right] \geq \mathbb{E}_{\mathbf{y}|\mathbf{t} \sim p(\mathbf{y}|\mathbf{t})} \left[ \log q(\mathbf{y}|\mathbf{t}) \right]
$$

and

$$
\begin{aligned}
\mathrm{I}(\mathbf{t}, \mathbf{y}) &\geq \mathbb{E}_{(\mathbf{t}, \mathbf{y}) \sim p(\mathbf{t}, \mathbf{y})} \left[ \log q(\mathbf{y}|\mathbf{t}) \right] \\
&= \mathbb{E}_{\mathbf{x} \sim p(\mathbf{x})} \mathbb{E}_{\mathbf{y}|\mathbf{x} \sim p(\mathbf{y}|\mathbf{x})} \mathbb{E}_{\mathbf{t}|\mathbf{x}, \mathbf{y} \sim p(\mathbf{t}|\mathbf{x}, \mathbf{y})} \left[ \log q(\mathbf{y}|\mathbf{t}) \right] \\
&= \mathbb{E}_{\mathbf{x} \sim p(\mathbf{x})} \mathbb{E}_{\mathbf{y}|\mathbf{x} \sim p(\mathbf{y}|\mathbf{x})} \mathbb{E}_{\mathbf{t}|\mathbf{x} \sim p(\mathbf{t}|\mathbf{x})} \left[ \log q(\mathbf{y}|\mathbf{t}) \right]
\end{aligned}
$$

where $p(\mathbf{t}|\mathbf{x}, \mathbf{y}) = p(\mathbf{t}|\mathbf{x})$ by the Markov chain assumption $\mathbf{y} \leftrightarrow \mathbf{x} \leftrightarrow \mathbf{t}$.

(III) VARIATIONAL LOWER BOUND FOR $\mathrm{I}(\mathbf{t}, \mathbf{y}) - \beta \mathrm{I}(\mathbf{x}, \mathbf{t})$

In summary, we have the following variational bounds for each term.

$$
\begin{aligned}
\mathrm{I}(\mathbf{t}, \mathbf{y}) &\geq \mathbb{E}_{\mathbf{x} \sim p(\mathbf{x})} \mathbb{E}_{\mathbf{y}|\mathbf{x} \sim p(\mathbf{y}|\mathbf{x})} \mathbb{E}_{\mathbf{t}|\mathbf{x} \sim p(\mathbf{t}|\mathbf{x})} \left[ \log q(\mathbf{y}|\mathbf{t}) \right] \\
\mathrm{I}(\mathbf{x}, \mathbf{t}) &\leq \mathbb{E}_{\mathbf{x} \sim p(\mathbf{x})} D_{\mathrm{KL}}(p(\mathbf{z}|\mathbf{x}), r(\mathbf{z})) + C
\end{aligned}
$$

which result in

$$
\begin{aligned}
&\mathrm{I}(\mathbf{t}, \mathbf{y}) - \beta \, \mathrm{I}(\mathbf{x}, \mathbf{t}) \\
&\geq \mathbb{E}_{\mathbf{x} \sim p(\mathbf{x})} \mathbb{E}_{\mathbf{y}|\mathbf{x} \sim p(\mathbf{y}|\mathbf{x})} \mathbb{E}_{\mathbf{t}|\mathbf{x} \sim p(\mathbf{t}|\mathbf{x})} \left[ \log q(\mathbf{y}|\mathbf{t}) \right] - \beta \, \mathbb{E}_{\mathbf{x} \sim p(\mathbf{x})} D_{\mathrm{KL}}(p(\mathbf{z}|\mathbf{x}), r(\mathbf{z})) + \mathrm{Const}.
\end{aligned}
$$

With proper choices of $r(\mathbf{z})$ and $p(\mathbf{z}|\mathbf{x})$, we assume that the Kullback-Leibler divergence $D_{\mathrm{KL}}(p(\mathbf{z}|\mathbf{x}), r(\mathbf{z}))$ is integrated analytically. We use the empirical data distribution to approximate $p(\mathbf{x}, \mathbf{y}) = p(\mathbf{x}) p(\mathbf{y}|\mathbf{x})$ and $p(\mathbf{x})$.

Our variational approximation is similar to the one from Alemi et al. (2017), which first developed variational lower bound on the information bottleneck for deep neural networks. However, our information bottleneck is different from theirs: their information bottleneck is the stochastic encoding $\mathbf{z}$ of the input $\mathbf{x}$ as itself, whereas our information bottleneck is a pairwise product of the stochastic encoding $\mathbf{z}$ and the input $\mathbf{x}$ where $\mathbf{z}$ is a Boolean random vector. Due to this difference, we appproximate $\mathrm{I}(\mathbf{x}, \mathbf{t})$ by using the lower bound of $\mathrm{I}(\mathbf{x}, \mathbf{z})$, instead of directly deriving the lower bound of $\mathrm{I}(\mathbf{x}, \mathbf{t})$.

# B    EXPERIMENTAL DETAILS

The settings of hyperparameter tuning include the followings (bold indicate the choice for our final model): the temperature for Gumbel-softmax approximation $\tau - \{0.1, 0.2, 0.5, \mathbf{0.7}, 1\}$, learning rate $- 5 \times 10^{-3}, 10^{-3}, 5 \times 10^{-4}, \mathbf{10^{-4}}, 5 \times 10^{-5}\}$ and $\beta - \{0, 0.001, \mathbf{0.01}, 0.1, 1, 10, 100\}$. We use Adam algorithm (Kingma & Ba, 2014) with batch size 100 for MNIST and 50 for IMDB, the coefficients used for computing running averages of gradient and its square $(\beta_1, \beta_2) = (0.5, 0.999)$, and $\epsilon = 10^{-8}$. We tuned the hyperparameters via grid search and picked up the hyperparameters that yield the best fidelity score on the validation set. All implementation is performed via PyTorch an open source deep learning platform (Paszke et al., 2017).

Approximator fidelity is prediction performance of the approximator that takes relaxation $\mathbf{t}^*$ as an input and black-box output as a target. In detail, we get a $\mathbf{t}^*$ with the Gumble-softmax sampling and make a prediction based on $\mathbf{t}^*$. This procedure is repeated for 12 times and the final prediction is made by averaging the 12 prediction scores. Rationale fidelity is prediction performance of the approximator that takes $\mathbf{t}$ as an input and black-box output as a target. (Note that $\mathbf{t}$ is a masked input that only takes the top $k$ chunks and set others to zero.) The final prediction is made by its prediction score. We use the prior $r_j(\mathbf{z}_j) = (k/J)^{\mathbf{z}_j}(1 - k/J)^{1-\mathbf{z}_j}$ for all experiments where $J$ is the total number of chunks and $r(\mathbf{z}) = \prod_j r_j(\mathbf{z}_j)$. The analytical form of the Kullback-Leibler divergence term is $D_{\mathrm{KL}}(p(\mathbf{z}|\mathbf{x}), r(\mathbf{z})) = \sum_{j=1}^{J} D_{\mathrm{KL}}(p(\mathbf{z}_j|\mathbf{x}), r_j(\mathbf{z}_j)) = \sum_{j=1}^{J} \sum_{z_j=0}^{1} p(\mathbf{z}_j|\mathbf{x}) (\log p(\mathbf{z}_j|\mathbf{x}) - \log r_j(\mathbf{z}_j))$.

## B.1    LSTM MOVIE SENTIMENT PREDICTION MODEL USING IMDB

**Black-Box Model Structure.** Each review is padded or cut to contain 15 sentences and 50 words for each sentence. The architecture consists of a word-embedding layer with size 50 for each word followed by two bidirectional LSTMs, a fully connected layer with two units, and a soft-max layer. The first LSTM layer encodes the word embedding vector and generates a word-representation vector with size 100 for each word. Within each sentence, the word representation vectors are elementwisely averaged to form a size 100 sentence representation vector. The second LSTM layer encodes the sentence representation vector and generates a size 60 review embedding vector.

**VIBI Structure.** We parameterize the explainer with a bidirectional LSTM and approximator with a 2D CNN. For the explainer, we use a bidirectional layer that returns multiple output vectors, each of which corresponds to a recurrent unit. Each element in the output vectors are averaged over all units, and then the averaged output vector is followed by log-softmax calculation. As a result, the explainer returns a vector of log-probabilities, each of which indicates whether or not each cognitive chunk will be selected as an input to the approximator. For the approximator, we use a convolutional layer followed by a ReLU activation function and max-pooling layer and a fully connected layer returning a size-2 vector followed by a log-softmax calculation. The final layer returns a vector of log-probabilities for the two sentiments (positive/negative).

## B.2    CNN DIGIT RECOGNITION MODEL USING MNIST

**Black-Box Model Structure.** The architecture consists of two convolutional layers with the kernel size 5 followed by a max-pooling layer with the pool size 2, two fully connected layers and a soft-max layer. The two convolutional layers contain 10 and 20 filters respectively and the two fully connected layers are composed of 50 and 10 units respectively.

**VIBI Structure.** We parameterize each the explainer and approximator using 2D CNNs. Structure of the explainer differs depending on the chunk size. For example, when $4 \times 4$ cognitive chunk is used, we use two convolutional layers with the kernel size 5 followed by a ReLU activation function and max-pooling layer with the pool size 2, and one convolutional layer with kernel size 1 returning a $7 \times 7$ 2D matrix followed by a log-softmax calculation. The final layer returns a vector

of log-probabilities for the 49 chunks. The three convolutional layers contains 8, 16, and 1 filters respectively. The output from the explainer indicates which cognitive chunks should be taken as an input for the approximator. We parameterize the approximator using two convolutional layers with kernel size 5 followed by a ReLU activation function and max-pooling layer with pool size 2 and with 32 and 64 filters respectively, and one fully connected layer returning a size-10 vector followed by a log-softmax calculation so that the final layer returns a vector of log-probabilities for the ten digits.

### B.3 TCR TO EPITOPE BINDING PREDICTION MODEL USING VDJDB AND IEDB

**Data.** We use two public datasets: VDJdb and IEDB. VDJdb (Shugay et al., 2017) contains the T-cell receptor (TCR) sequences with known antigen specificity (i.e., epitope sequences). We use a VDJdb dataset preprocessed by Jokinen et al. (2019) (See their paper for details in data preprocessing). The preprocessed dataset contains 5,784 samples (2,892 positively and 2,892 negatively binding pairs of TCR and epitope). This dataset consists of 4,363 unique TCR sequences and 21 unique epitope sequences. We group the pairs of TCR and epitope sequences into training, validation, and test sets, which have 4,627, 578, and 579 pairs, respectively. IEDB (Vita et al., 2014) contains TCR sequences and corresponding epitopes. We used an IEDB dataset preprocessed by Jurtz et al. (2018). The preprocessed dataset contains 9,328 samples (positively binding pairs only) and consists of 9,221 unique TCR sequences and 98 unique epitopes sequences. We used the IEDB dataset as out-ouf-samples. There are 6 epitopes contained in both VDJdb and IEDB dataset: `GILGFVFTL`, `GLCTLVAML`, `NLVPMVATV`, `LLWNGPMAV`, `YVLDHLIVV`, `CINGVCWTV`.

**Black-Box Model Structure.** Each TCR and epitope sequence is padded or cut to contain 20 and 13 amino acids, respectively. The embedding matrix has a size 24 and is initialized with BLOSUM50 matrix (Henikoff & Henikoff, 1992). The architecture consists of two sequence encoders that process TCR and epitope sequences each, and three dense layers that process the encoded sequences together. The TCR encoder consists of a dropout layers with the drop-out probability 0.3, two convolutional layers with the kernel size 3 followed by a batch normalization layer, a ReLU activation function and max-pooling layer with the pool size 3. The two convolutional layers contain 32 and 16 filters respectively. Structure of the epitope encoder is the same with the one from the TCR encoder. We optimized the models with the following search space (bold indicate the choice for our final model): the batch size – $\{25, \mathbf{50}, 100\}$, learning rate – $0.05, 0.01, 0.005, \mathbf{0.001}, 0.0005\}$ and filter size – $\{(16, 8), (\mathbf{32}, \mathbf{16})\}$.

**VIBI Structure.** We parameterize the explainer and approximator using 2D CNNs and several dense layers. Each TCR and epitope sequence is preprocessed and embedded in the same way as for the black-box model. We have two types of explainers: one for TCR sequence and another for epitope sequence. Each explainer encodes both TCR and epitope sequences and concatenates them through three dense layers followed by a ReLU activation function. For the TCR explainer, the three dense layers contain 32, 16, 20 hidden-units, respectively. For the epitope explainer, the three dense layers contain 32, 16, 13 hidden-units, respectively. The TCR and epitope encoders have the same architectures with those of the black-box model. Each explainer then returns a vector of log-probabilities that indicate which peptides in TCR or epitope should be selected as an explanation. The approximator has the same architecture as the black-box model. We optimize the models with the following search space (bold indicate the choice for our final model): the batch size – $\{25, \mathbf{50}, 100\}$, learning rate – $0.0001, \mathbf{0.001}, 0.01, 0.1\}$.

### B.4 BASELINE METHODS

**Hyperparameter tuning.** For L2X, we used the same hyperparameter search space of VIBI. For LIME, we tuned the segmentation filter size over $\{1, \mathbf{2}, 4\}$ for MNIST and $\{\mathbf{10}, 25\}$ for IMDB. For all methods, we tuned the hyperparameters via grid search and picked up the hyperparameters that yield the best fidelity score on the validation set.

**Cognitive chunk.** LIME, SHAP and Saliency yield an attribution score for each feature. A chunk-attribution score is an average of (absolute) attribution scores of features that belong to the chunk. Top $k$ chunks that have the highest chunk-attribution scores are selected. L2X selects chunks in the same way as VIBI.

**Fidelity evaluation.** LIME, SHAP, Saliency, and L2X all have their own approximators. LIME and SHAP have a sparse linear approximator for each black-box instance (See Section 3.4 in Ribeiro et al. (2016), and Equation (3) in Lundberg & Lee (2017)). Saliency has a 1st-order Taylor approximation to each black-box instance (See Equation (3) in Simonyan et al. (2013)). The approximator fidelity of LIME, SHAP, and Saliency is calculated using their proposed approximators that are composed of the selected features. L2X has a deep neural network that approximates a black-box model (See Section 4 in Chen et al. (2018)). The approximator and rationale fidelity of L2X is calculated in the same way as VIBI.

# C  INTERPRETABILITY EVALUATED BY HUMANS

## C.1  LSTM MOVIE SENTIMENT PREDICTION MODEL USING IMDB DATASET

EVALUATION

Table 3: Evaluation of Interpretability on an LSTM movie sentiment prediction model using IMDB.

| Black-Box Output | Recognized by Mturk worker | Saliency | LIME | L2X | VIBI (Ours) |
|---|---|---|---|---|---|
| Positive | Positive | 19.8 | 17.4 | 17.6 | **24.3** |
| Positive | Negative | 12.6 | **6.8** | 7.2 | 16.9 |
| Positive | Neutral | 25.6 | 18.8 | 24.2 | **11.1** |
| Negative | Positive | 11.0 | **10.6** | 11.6 | 16.2 |
| Negative | Negative | 14.4 | 16.4 | 18.0 | **20.4** |
| Negative | Neutral | 16.6 | 30.0 | 21.4 | **11.2** |

The percentage of samples belongs to each combination of the black-box output and the sentiment recognized by workers at Amazon Mechanical Turk (`https://www.mturk.com/`) are showed

We evaluated interpretability of the methods on the LSTM movie sentiment prediction model. The interpretable machine learning methods were evaluated by workers at Amazon Mechanical Turk (`https://www.mturk.com/`) who are awarded the Masters Qualification (i.e. high performance workers who have demonstrated excellence across a wide range of task). Randomly selected instances (200 for VIBI and 100 for the others) were evaluated for each method. 5 workers are assigned per instance.

We provided instances that the black-box model had correctly predicted and asked humans to infer the output of the primary sentiment of the movie review (Positive/Negative/Neutral) given five key words selected by each method. See the survey example below for further details. Note that this is a proxy for measuring how well humans infer the black-box output given explanations; we use such proxy because the workers are general public who are not familiar with the term 'black-box' or 'output of the model.'

In Table 3, the percentage of samples belongs to each combination of the black-box output and the sentiment recognized by the workers are showed. VIBI has the highest percentage of samples belonging to the Positive/Positive or Negative/Negataive and the lowest percentage of samples belonging to the Positive/Neutral or Negative/Neutral. LIME has the lowest percentage of samples belonging to the Positive/Negative or Negative/Positive, but it is because LIME tends to select words such as 'that', 'the', 'is' so that most of samples are recognized as Neutral.

SURVEY EXAMPLE FOR IMDB

Title: Label sentiment given a few words.
Description: Recognize the primary sentiment of the movie review given a few words only.

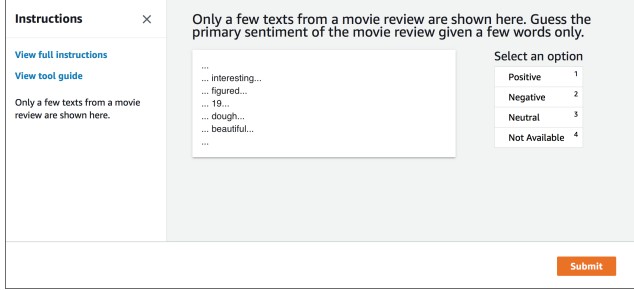

Figure 5: A survey example of MTurk evaluation on the LSTM movie sentiment prediction model

## C.2  CNN DIGIT RECOGNITION MODEL USING MNIST

EVALUATION

Table 4: Evaluation of Interpretability on a 2d CNN digit recognition model using MNIST.

| MNIST Digit | Saliency | LIME | L2X | VIBI (Ours) |
|---|---|---|---|---|
| 0 | **3.200** | 2.000 | 2.333 | 3.000 |
| 1 | **4.393** | 0.795 | 1.263 | 3.913 |
| 2 | 3.125 | 1.200 | 1.400 | **3.200** |
| 3 | 3.286 | 1.833 | 2.429 | **3.625** |
| 4 | 3.333 | 1.000 | 1.857 | **3.857** |
| 5 | **3.167** | 1.381 | 2.000 | 2.875 |
| 6 | 3.333 | 1.000 | 1.889 | **3.625** |
| 7 | 3.667 | 2.000 | 1.667 | **4.000** |
| 8 | **3.750** | 1.333 | 2.667 | 3.500 |
| 9 | 3.222 | 1.143 | 1.857 | **3.667** |
| Ave. over digits | 3.448 | 1.369 | 1.936 | **3.526** |

The average scores (0–5 scale) evaluated by graduate students at XXX University (Blinded due to the double blinding reviewing) are showed.

We evaluated interpretability of the methods on the CNN digit recognition model. The interpretable machine learning methods were evaluated by 16 graduate students at XXX University (Blinded due to the double blinding reviewing) who have taken at least one graduate-level machine learning class. Randomly selected 100 instances were evaluated for each method. On average, 4.26 students are assigned per instance. See the survey example below for further details. For the digit recognition model, we asked humans to directly score the explanation on a 0–5 scale.

In Table 4, the average score per digit is showed. VIBI outperforms L2X and LIME and slightly outperforms Saliency in terms of the average score over digits. VIBI outperforms at digit 2, 3, 4, 6, 7, and 9, and performs comparable to Saliency at 0, 1, 5, 8.

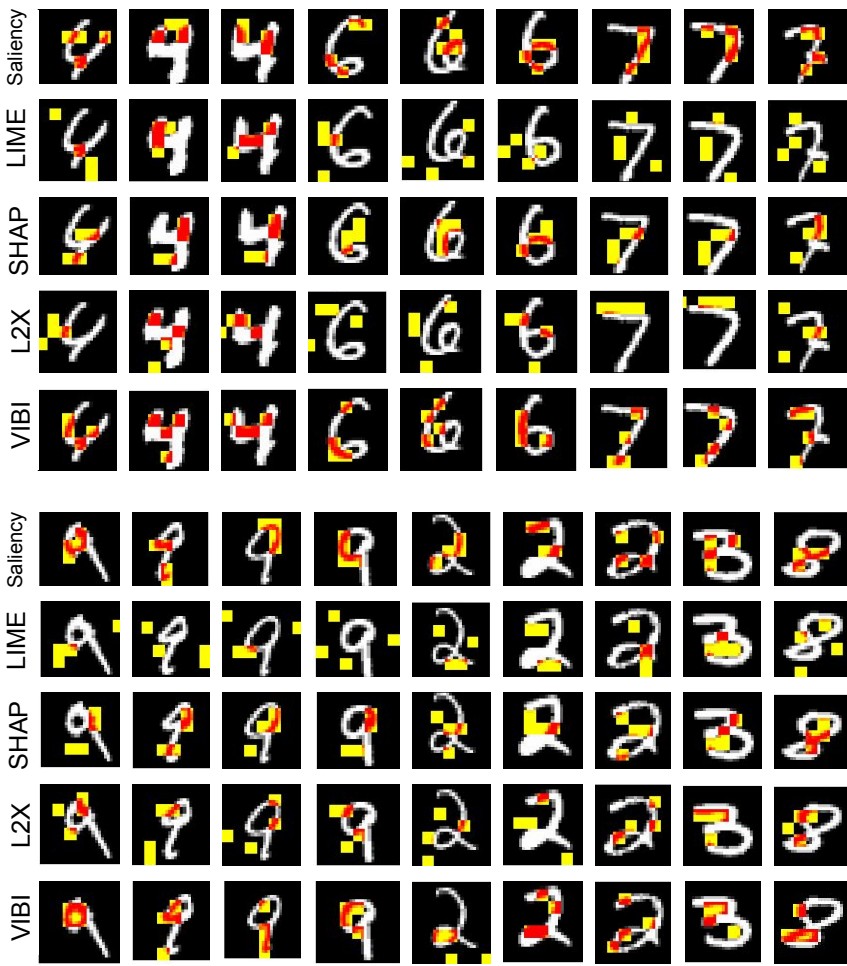

Figure 6: The hand-written digits and explanations provided by VIBI and the baselines. The examples are randomly selected from the validation set. The selected patches are colored red if the pixel is activated (i.e. white) and yellow otherwise (i.e. black). A patch composed of $4 \times 4$ pixels is used as a cognitive chunk and $k = 4$ patches are identified for each image.

# Instruction

- 2D CNN model is used for digit recognition for MNIST dataset

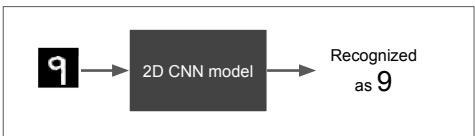

- Now, an interpretable learning method explains a decision made by the 2D CNN model:

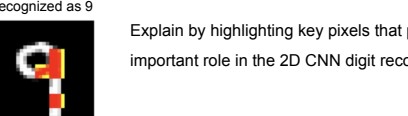

MNIST is a large dataset contains 28 x 28 sized images of handwritten digits (0 to 9). Here, a 2D convolutional neural network (CNN) is used for the digit recognition for MNIST. Several interpretable machine learning methods are learned to explain the model by highlighting key pixels that play an important role in the CNN digit recognition. The highlighted pixels provides an explanation for a handwritten image why the CNN model recognized the handwriting as it does. Your task is to evaluate the explanation for each instance on a scale 0 to 5.

## Please score each instance based on following criteria:

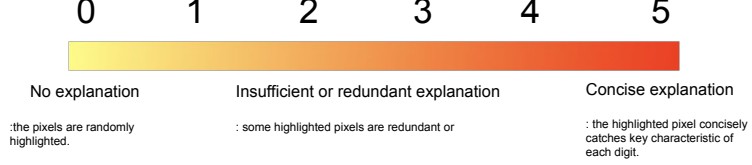

Figure 7: A survey example of evaluation on the MNIST digit recognition model.

# D FIDELITY

Table 5: Evaluation of rationale fidelity on LSTM movie sentiment prediction model using IMDB.

|  |  |  | L2X | VIBI (Ours) |  |  |  |  |  |
|---|---|---|---|---|---|---|---|---|---|
|  | chunk size | k | 0 | 0.001 | 0.01 | 0.1 | 1 | 10 | 100 |
| Accuracy | sentence | 1 | 0.727 | 0.693 | 0.711 | 0.731 | 0.729 | **0.734** | **0.734** |
|  | word | 5 | 0.638 | 0.657 | **0.666** | 0.657 | 0.648 | 0.640 | 0.654 |
|  | 5 words | 1 | 0.601 | 0.630 | 0.624 | **0.632** | 0.628 | 0.623 | 0.628 |
|  | 5 words | 3 | **0.694** | 0.660 | 0.662 | 0.660 | 0.662 | 0.660 | 0.660 |
| F1-score | sentence | 1 | 0.581 | 0.547 | 0.562 | 0.567 | 0.585 | **0.586** | **0.586** |
|  | word | 5 | 0.486 | 0.521 | **0.551** | 0.512 | 0.516 | 0.508 | 0.526 |
|  | 5 words | 1 | 0.478 | 0.506 | 0.500 | **0.540** | 0.501 | 0.498 | 0.504 |
|  | 5 words | 3 | **0.551** | 0.528 | 0.529 | 0.522 | 0.529 | 0.528 | 0.525 |

Rationale fidelity quantifies ability of the selected chunks to infer the black-box output. A large rationale fidelity implies that the selected chunks account for a large portion of the approximator fidelity. Prediction accuracy and F1-score of the approximator for the CNN model are shown. ($\beta = 0, 0.001, 0.01, 0.1, 1, 10, 100$)

Table 6: Evaluation of approximator fidelity on LSTM movie sentiment prediction model using IMDB.

|  |  |  | Saliency | LIME | SHAP | L2X | VIBI (Ours) |  |  |  |  |  |  |
|---|---|---|---|---|---|---|---|---|---|---|---|---|---|
|  | chunk size | k |  |  |  | 0 | 0.001 | 0.01 | 0.1 | 1 | 10 | 100 |
| Accuracy | sentence | 1 | 0.387 | 0.727 | 0.495 | 0.876 | 0.877 | 0.869 | 0.877 | 0.879 | 0.879 | **0.884** |
|  | word | 5 | 0.419 | 0.756 | 0.501 | 0.738 | 0.766 | 0.772 | 0.744 | **0.773** | 0.763 | 0.767 |
|  | 5 words | 1 | 0.424 | 0.290 | 0.496 | 0.759 | **0.784** | 0.780 | 0.764 | 0.774 | 0.778 | 0.774 |
|  | 5 words | 3 | 0.414 | 0.679 | 0.491 | 0.833 | **0.836** | 0.831 | 0.835 | 0.834 | 0.830 | 0.833 |
| F1-score | sentence | 1 | 0.331 | 0.564 | 0.400 | 0.721 | 0.693 | 0.707 | 0.730 | 0.730 | 0.727 | **0.734** |
|  | word | 5 | 0.350 | 0.585 | 0.413 | 0.565 | 0.607 | 0.616 | 0.594 | **0.620** | 0.609 | 0.612 |
|  | 5 words | 1 | 0.360 | 0.302 | 0.418 | 0.621 | **0.641** | 0.622 | 0.624 | 0.615 | 0.622 | 0.616 |
|  | 5 words | 3 | 0.352 | 0.523 | 0.409 | 0.680 | **0.683** | 0.674 | 0.681 | 0.677 | 0.669 | 0.682 |

Approximator fidelity quantifies ability of the approximator to imitate the behaviour of a black-box. Prediction accuracy and F1-score of the approximator for the LSTM model are shown. ($\beta = 0, 0.001, 0.01, 0.1, 1, 10, 100$)

Table 7: Evaluation of the rationale fidelity on CNN digit recognition model using MNIST.

| | | | L2X | VIBI (Ours) | | | | | |
| | chunk size | k | 0 | 0.001 | 0.01 | 0.1 | 1 | 10 | 100 |
|---|---|---|---|---|---|---|---|---|---|
| | $1 \times 1$ | 64 | 0.694 | 0.690 | 0.726 | 0.689 | 0.742 | 0.729 | **0.766** |
| | $1 \times 1$ | 96 | 0.814 | 0.831 | 0.780 | 0.806 | **0.859** | 0.765 | 0.826 |
| | $1 \times 1$ | 160 | 0.903 | 0.907 | 0.905 | 0.917 | 0.917 | **0.928** | 0.902 |
| | $2 \times 2$ | 16 | 0.735 | **0.795** | 0.750 | 0.771 | 0.732 | 0.753 | 0.769 |
| | $2 \times 2$ | 24 | 0.776 | 0.855 | 0.834 | 0.856 | **0.868** | 0.854 | 0.847 |
| Accuracy | $2 \times 2$ | 40 | 0.811 | 0.914 | 0.914 | 0.915 | 0.903 | 0.918 | **0.935** |
| | $2 \times 2$ | 80 | 0.905 | 0.949 | 0.940 | 0.939 | **0.962** | 0.941 | 0.923 |
| | $4 \times 4$ | 4 | 0.650 | 0.655 | 0.650 | **0.775** | 0.717 | 0.682 | 0.681 |
| | $4 \times 4$ | 6 | 0.511 | **0.858** | 0.706 | 0.701 | 0.708 | 0.690 | 0.730 |
| | $4 \times 4$ | 10 | 0.835 | 0.835 | 0.824 | **0.933** | 0.875 | 0.854 | 0.782 |
| | $4 \times 4$ | 20 | 0.954 | **0.962** | 0.815 | 0.934 | 0.929 | 0.946 | 0.943 |
| | $1 \times 1$ | 64 | 0.684 | 0.679 | 0.716 | 0.670 | 0.734 | 0.710 | **0.755** |
| | $1 \times 1$ | 96 | 0.808 | 0.825 | 0.750 | 0.803 | **0.854** | 0.750 | 0.820 |
| | $1 \times 1$ | 160 | 0.898 | 0.902 | 0.899 | 0.912 | 0.913 | **0.924** | 0.897 |
| | $2 \times 2$ | 16 | 0.720 | **0.786** | 0.738 | 0.761 | 0.723 | 0.744 | 0.769 |
| | $2 \times 2$ | 24 | 0.766 | 0.848 | 0.836 | 0.851 | 0.858 | **0.859** | 0.840 |
| F1-score | $2 \times 2$ | 40 | 0.798 | 0.914 | 0.910 | 0.910 | 0.898 | 0.914 | **0.931** |
| | $2 \times 2$ | 80 | 0.901 | 0.946 | 0.936 | 0.930 | **0.959** | 0.938 | 0.918 |
| | $4 \times 4$ | 4 | 0.634 | 0.658 | 0.637 | **0.763** | 0.704 | 0.671 | 0.669 |
| | $4 \times 4$ | 6 | 0.493 | **0.852** | 0.693 | 0.687 | 0.692 | 0.675 | 0.720 |
| | $4 \times 4$ | 10 | 0.828 | 0.827 | 0.816 | **0.928** | 0.869 | 0.849 | 0.773 |
| | $4 \times 4$ | 20 | 0.950 | 0.959 | 0.806 | 0.931 | 0.926 | 0.942 | 0.940 |

Rationale fidelity quantifies ability of the selected chunks to infer the black-box output. A large rationale fidelity implies that the selected chunks account for a large portion of the approximator fidelity. Prediction accuracy and F1-score of the approximator for the CNN model are shown. ($\beta = 0, 0.001, 0.01, 0.1, 1, 10, 100$)

Table 8: Evaluation of the approximator fidelity on CNN digit recognition model using MNIST.

| | | | Saliency | LIME | SHAP | L2X | VIBI (Ours) | | | | | |
| | chunk size | k | | | | 0 | 0.001 | 0.01 | 0.1 | 1 | 10 | 100 |
|---|---|---|---|---|---|---|---|---|---|---|---|---|
| | $1 \times 1$ | 64 | 0.944 | **0.982** | 0.955 | 0.933 | 0.959 | 0.962 | 0.959 | 0.960 | 0.952 | 0.953 |
| | $1 \times 1$ | 96 | 0.956 | **0.986** | 0.950 | 0.963 | 0.963 | 0.951 | 0.967 | 0.968 | 0.953 | 0.962 |
| | $1 \times 1$ | 160 | 0.964 | **0.989** | 0.958 | 0.970 | 0.967 | 0.973 | 0.974 | 0.974 | 0.974 | 0.967 |
| | $2 \times 2$ | 16 | 0.912 | 0.770 | 0.942 | 0.934 | 0.945 | 0.941 | **0.948** | 0.938 | 0.939 | 0.940 |
| | $2 \times 2$ | 24 | 0.938 | 0.807 | 0.954 | 0.951 | 0.956 | 0.955 | 0.953 | 0.953 | 0.953 | **0.960** |
| Accuracy | $2 \times 2$ | 40 | 0.957 | 0.859 | 0.954 | **0.967** | 0.965 | 0.966 | 0.962 | **0.967** | 0.965 | **0.967** |
| | $2 \times 2$ | 80 | 0.966 | 0.897 | 0.966 | 0.976 | **0.977** | 0.974 | 0.972 | **0.977** | 0.971 | 0.973 |
| | $4 \times 4$ | 4 | 0.863 | 0.609 | 0.948 | **0.953** | 0.922 | 0.928 | 0.948 | 0.942 | 0.942 | **0.953** |
| | $4 \times 4$ | 6 | 0.906 | 0.637 | 0.936 | 0.957 | **0.963** | 0.954 | 0.956 | 0.953 | **0.963** | 0.962 |
| | $4 \times 4$ | 10 | 0.949 | 0.705 | 0.951 | 0.965 | **0.971** | 0.959 | 0.967 | 0.961 | 0.969 | 0.964 |
| | $4 \times 4$ | 20 | 0.963 | 0.771 | 0.955 | 0.974 | **0.977** | 0.975 | 0.975 | 0.973 | 0.975 | 0.974 |
| | $1 \times 1$ | 64 | 0.938 | **0.981** | 0.952 | 0.930 | 0.956 | 0.960 | 0.956 | 0.957 | 0.950 | 0.950 |
| | $1 \times 1$ | 96 | 0.950 | **0.985** | 0.948 | 0.961 | 0.961 | 0.954 | 0.965 | 0.966 | 0.951 | 0.960 |
| | $1 \times 1$ | 160 | 0.959 | **0.989** | 0.954 | 0.969 | 0.965 | 0.971 | 0.973 | 0.972 | 0.972 | 0.966 |
| | $2 \times 2$ | 16 | 0.902 | 0.755 | 0.938 | 0.930 | 0.942 | 0.936 | **0.944** | 0.934 | 0.936 | 0.936 |
| | $2 \times 2$ | 24 | 0.932 | 0.795 | 0.951 | 0.949 | 0.954 | 0.952 | 0.950 | 0.951 | 0.950 | **0.958** |
| F1-score | $2 \times 2$ | 40 | 0.952 | 0.853 | 0.946 | **0.965** | 0.963 | 0.964 | 0.961 | **0.965** | 0.963 | **0.965** |
| | $2 \times 2$ | 80 | 0.962 | 0.892 | 0.963 | 0.974 | **0.976** | 0.973 | 0.972 | 0.975 | 0.969 | 0.971 |
| | $4 \times 4$ | 4 | 0.849 | 0.588 | 0.943 | **0.951** | 0.917 | 0.923 | 0.944 | 0.938 | 0.939 | 0.950 |
| | $4 \times 4$ | 6 | 0.895 | 0.621 | 0.920 | 0.954 | **0.961** | 0.951 | 0.953 | 0.950 | 0.960 | 0.959 |
| | $4 \times 4$ | 10 | 0.943 | 0.689 | 0.946 | 0.963 | 0.969 | 0.956 | 0.965 | 0.958 | **0.968** | 0.961 |
| | $4 \times 4$ | 20 | 0.959 | 0.763 | 0.952 | 0.972 | **0.976** | 0.972 | 0.974 | 0.972 | 0.973 | 0.972 |

Approximator fidelity quantifies ability of the approximator to imitate the behaviour of a black-box. Prediction accuracy and F1-score of the approximator for the CNN model are shown. ($\beta = 0, 0.001, 0.01, 0.1, 1, 10, 100$)

# E EXTRA EXPERIMENTS

## E.1 CHOICE OF $k$

The compressiveness of explanations depends on the sparsity $k$ (i.e., the number of cognitive chunks to be selected). A larger $k$ allows the information bottleneck to convey more information about output, but it gives less compressive explanations than a smaller $k$. For deciding $k$, we recommend choosing the minimum possible $k$ that achieves a target fidelity because an unnecessarily large $k$ can make redundant explanations. Figure 8 shows how VIBI works under different sparsity $k$. When we increase $k$, VIBI tends to select chunks that are the same or nearby the previously selected chunks and additionally select new chunks that catch another characteristics of digits. The characteristics that are caught at $k = 4$ tend to be caught again at a larger $k$s.

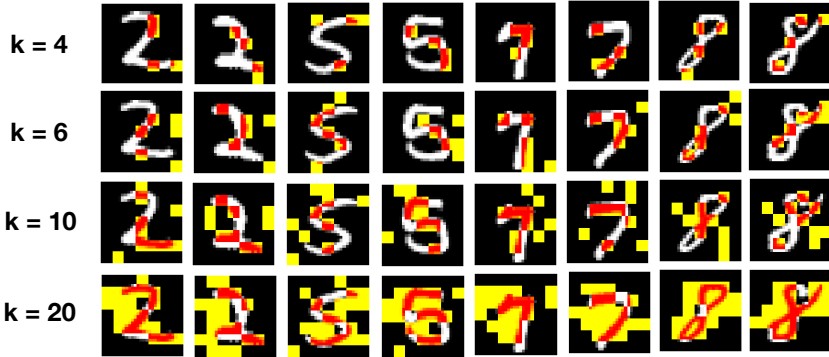

Figure 8: The hand-written digits and explanations provided by VIBI using various number of chunks ($k$). The examples are randomly selected from the validation set. The selected patches are colored red if the pixel is activated (i.e. white) and yellow otherwise (i.e. black). A patch composed of $4 \times 4$ pixels is used as a cognitive chunk and $k = 4, 6, 10, 20$ patches are identified for each image.

## E.2 CHOICE OF CHUNK SIZE

The chunk size changes the way the information is represented in the information bottleneck, which can affect interpretability. For choosing the chunk size, a greater fidelity does not always mean better interpretability. For example, in the MNIST experiment, the models with the chunk size $1 \times 1$ have achieved better fidelity than those with $2 \times 2$ and $4 \times 4$ under the same number of pixels that are used for the information bottleneck, but $1 \times 1$ is less recognizable than $2 \times 2$ or $4 \times 4$ (Figure 9).

Choosing the chunk size is not trivial, and the strategy varies for different domains (except the case that each (raw) feature itself is cognitive (e.g., age, gender) so that the chunk size should be 1). Obviously, the chunk size should be larger than the size of the smallest cognitive unit. For example, in the IMDB experiment, the smallest cognitive unit was set to 50 because 50 raw features compose a word. In our sentiment prediction task, using a word as the chunk size can be enough to explain the decision (i.e., positive/negative sentiment); to infer the sentiment, we do not need to know a whole sentence. However, we may need to use a phrase or sentence as the chunk size for complex tasks such as disease diagnosis or patient need detection since decisions on such tasks are made based on contextual information. In the MNIST experiment, we set the chunk size to be $4 \times 4$ because the

digits' stroke width is about to 4. However, we may need to use different chunk sizes, for example, for automated visual inspection for identifying defects on a factory production line. In this case, the chunk size can be set to the average defect size on the production line.

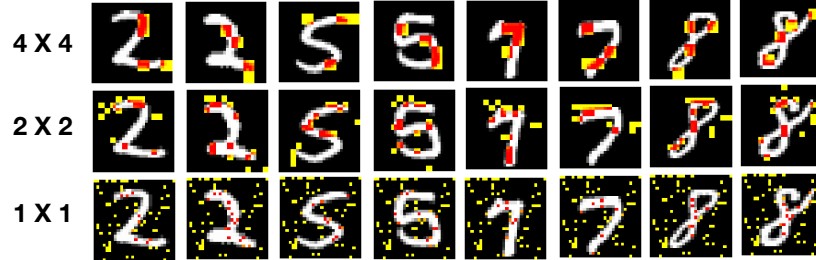

Figure 9: The hand-written digits and explanations provided by VIBI using various chunk sizes. The same examples with Figure 8 are shown. The selected patches are colored red if the pixel is activated (i.e. white) and yellow otherwise (i.e. black). A patch composed of $4 \times 4$, $2 \times 2$, $1 \times 1$ pixels is used as a cognitive chunk.

### E.3 CHOICE OF APPROXIMATOR

Different approximators affect fidelity and interpretability of the explanation, which makes sense both intuitively and theoretically. In theory, explanation is obtained via solving the problem in Equation (2) where $I(\mathbf{t}, \mathbf{y})$ represents the approximator. Different models of the approximator result in different $I(\mathbf{t}, \mathbf{y})$ and optimizing the information bottleneck objective amounts to learning a decent approximator. Intuitively, different models of the approximator indeed affect the quality as well as evaluated fidelity of the explanation. If the model of approximator has low capacity, $I(\mathbf{t}, \mathbf{y})$ and fidelity tend to be low. To remedy this, explainer has to generate less brief explanations; vice versa.

In addition to the approximator capacity, the chunk sizes, more precisely chunk types, can be an essential factor for choosing approximators. In the IMDB experiment, we investigated how the two approximators affect the fidelity and interpretability. As seen in Table 9, the LSTM approximator has lower fidelity than the CNN approximator when we use a word for the chunk size, while both approximators achieve similar fidelity when we use a sentence for the chunk size. When we use a sentence as a cognitive chunk, the LSTM achieves better interpretability than CNN, which is more substantial than when we use a word as a cognitive chunk. This may indicate that a sentence (as an explanation) should be processed with an approximator that resembles the black-box model, while a group of single words is relatively free from such a constraint. This may be because a sentence is more similar to the original input (i.e., movie review – one or two paragraphs) than a group of single words.

Table 9: Fidelity and interpretability under different approximators.

| chunk size | k | Approximator Fidelity | | Rationale Fidelity | | Interpretability | |
|---|---|---|---|---|---|---|---|
| | | CNN | LSTM | CNN | LSTM | CNN | LSTM |
| sentence | 1 | $87.7 \pm 0.6$ | $87.8 \pm 1.0$ | $73.1 \pm 0.8$ | $71.7 \pm 2.4$ | 45.1 | 65.9 |
| word | 5 | $74.4 \pm 0.8$ | $61.3 \pm 3.0$ | $65.7 \pm 0.8$ | $57.1 \pm 2.4$ | 49.4 | 51.3 |

Prediction accuracy is reported for all measures. Randomely selected instances (100 for each) were evaluated for each model. 5 workers are assigned per instance.

In practice, we recommend choosing the approximator that matches the black-box model, especially if a cognitive chunk resembles the original input. We then recommend starting with an approximator with enough capacity and reducing it as far as the approximator achieves a target fidelity because a large capacity usually makes it hard to learn.

