# OpenReview forum: "Explaining A Black-box By Using A Deep Variational Information Bottleneck Approach"
_ICLR.cc/2020/Conference — Reject_

### Official Review · AnonReviewer2 · 2019-10-17
**Official Blind Review #2**

**Rating:** 6

**Review:**

[Due to the rebuttal, my score was raised from a weak reject to a weak accept]

Summary
The paper addresses the problem of interpreting predictions/decisions of a black-box classifier/regressor by masking the parts of the input that were most relevant. The proposed approach consists of, first, manually designing “cognitive chunks” of input data, e.g. individual words for sentiment classification or fixed-size image-patches for image-classification. Then, a variational IB framework is used to infer which of these chunks are relevant for the classifier’s decision. Additionally, there is a (hard) constraint, making sure that only a fixed (small) number of chunks is used. The bottleneck variable, in this case, is a sparse-chunk representation of the data. The latter is obviously a more compressed representation of the data, but importantly it is a more compressed representation that contains the largest possible amount of relevant information about the decision (because of properties of the information-bottleneck objective). Both factors together, according to the paper, constitute a “good” (i.e. brief but comprehensive) explanation which allows for interpretability and attribution of the black-box system’s decision. The method is evaluated on three tasks (sentiment prediction, image classification, TCR to epitope binding prediction) and performance is reported to be on-par or better than state-of-the-art methods.

Contributions
-) Application of the IB-method for generating summaries of decision-relevant input-data, which are good candidates for interpretability. The theoretical properties of the IB objective are appealing for producing interpretable data-summaries.
-) Adaptation of the variational IB framework, using bits and pieces reported in the literature such that the bottleneck variable is a sparse, binary vector over “cognitive chunks”.
-) Experimental evaluation, where human judges rate the “interpretability“ of various state-of-the-art attribution methods.

Quality, Clarity, Novelty, Impact
The paper addresses a timely and important problem, particularly the IB framework could add some solid theoretical footing (the “theory of relevant information”) to the field of interpetability methods. The paper is well written (though it needs another pass for typos, etc.), related methods and literature are discussed and compared against, and the specific variational IB objective is introduced nicely. Large parts of the method (deep variational IB, VI with categorical variables) have been published before, but these parts are combined in a novel and original way. My main issues with the current paper are (I) interpretability and comprehensiveness are not necessarily the same as maximum compression of maximally relevant information, (II) the method (in theory) depends strongly on the quality of the approximator, this is currently not mentioned and not explored, (III) the experimental section is currently not very strong, in particular the MNIST experiment. See more details for the main issues below. Overall, I personally think that the main idea of the paper is interesting, mature and fleshed out enough for a publication, however, the experimental section is somewhat lacking and (II) is missing from the current manuscript. While the method has theoretical advantages, empirically it seems to perform more or less equal to L2X (but the chunks produced seem qualitatively different which is interesting). I am therefore slightly leaning towards suggesting a major revision of the paper, but I am happy to be convinced otherwise by the other reviewers and the authors during discussion/rebuttal. (I would rate the paper as "borderline", but it seems that this year's review system only allows for "weak reject" or "weak accept", so I'll go for "weak reject" for now).

Improvements / major issues

(I) Good compression of highly relevant information is not (always) the same as good interpretability/comprehensiveness. In the limit, the bottleneck variable captures a minimal sufficient statistic, i.e. a maximally compressed version of all relevant information - for finite beta, the bottleneck approximates such a minimal sufficient statistic. From a theoretical point of view this is very appealing, since it is guaranteed to cover a maximal amount of information (given a certain level of compression). But the way this information is represented matters a lot for interpretability - any reversible mapping of the bottleneck variable does not change its information content but can have substantial effects on interpretability, e.g. consider encrypting or randomly perturbing elements of the explanation (i.e. the selected cognitive chunks). This is a major open problem, and some theoretical grounding in the IB framework helps by talking about this problem in very concrete terms. While I would not expect the paper to solve the problem in full generality, some discussion, and perhaps adding a “shortcomings” section would be nice.

(II) The relevance of information is measured via I(t;y), which ultimately boils down to the approximation q(y|t). The quality of this approximation is crucial, which can of course be seen by how it influences the tightness of the bound. While I appreciate that the paper investigates the quality of the approximation to some degree (by inspecting the approximator fidelity), I would highly appreciate a thorough discussion of this issue (because ultimately the method will produce cognitive chunks that are relevant for q(y|t), not p(y|x) - the interpretations can be trusted only if q matches fairly well). It would be very interesting to see how quickly interpretability degrades with lower-quality q(y|t) - the latter would of course require more experiments with human “interpreters” which I would not expect to be easily feasible within the rebuttal period.
Another interesting experiment to test the match between q(y|t) and p(y|x) would be to “minimally intervene” on the input-chunks suggested by the method and see whether that actually affects the predictions of the black-box models. E.g. do small random perturbations to the selected cognitive chunks in the MNIST digits change the prediction of the black-box classifier? Compare this against small random perturbations in arbitrary chunks of the input.

(III) Experimental section: I’m fairly happy with the IMDB experiment, and the TCR to Epitope binding is a nice non-standard application but I find the quality and significance of the results a bit hard to judge. My main concern though is the MNIST experiment: what I would have expected was the following: cognitive chunks are shown to participants and they need to guess the correct number (just like in the IMDB experiment). In the experiment reported in the paper, I’m afraid that there’s a certain bias for judges favoring explanations that lie on the digits rather than off digits. It remains unclear whether they simply prefer the chunks selected by VIBI over other methods, or whether they have actually gained more understanding of how the black-box makes decisions.

(IV) Table 2 can easily be misleading because entries with highest mean-accuracy are marked in bold, regardless of whether confidence intervals overlap with other entries or not. Please fix this by either only marking entries in bold where the error bars don’t overlap with an entry in the same row, or marking all entries in bold that lie within the error bars of the best-performing entry. Particularly for “Approximate Fidelity” VIBI often does not perform significantly better than L2X but performs roughly equally well. Of course it would also help to run more repetitions to potentially shrink confidence intervals.

(V) Please state the (parametric form) of the prior r(z*) used for the experiments. Also state the analytical expression for the KL-term in the final objective that this prior leads to.

Minor comments

a) Please add some discussion on how the method depends on hand-crafting cognitive chunks, and how hard/easy this might be for different domains.

b) Rather than fixing r(z), other papers have proposed to optimize the prior as well (typically in the context of VAEs / VIB) which is well justified from an IB perspective. It might be interesting to explore these possibilities for VIBI as well in the future.

[1] Fixing a broken ELBO. Alemi et al. 2017
[2] The beta-VAE’s Implicit Prior. Hoffman et al. 2017

c) Instead of fixing the number of cognitive chunks in advance, it could also be interesting to infer that number as well (as a future extension of the method). This could either be achieved via a sparsity-inducing prior r(z), or perhaps by borrowing some ideas from the Deterministic IB [3], and its variational version.

[3] The deterministic information bottleneck. Strouse and Schwab. 2016

**Experience Assessment:**

I have published one or two papers in this area.

**Review Assessment: Checking Correctness Of Derivations And Theory:**

I assessed the sensibility of the derivations and theory.

**Review Assessment: Checking Correctness Of Experiments:**

I assessed the sensibility of the experiments.

**Review Assessment: Thoroughness In Paper Reading:**

I read the paper at least twice and used my best judgement in assessing the paper.

---

> ### Author Response · Authors · 2019-11-09
> **Thank you for providing constructive comments and critiques.**
>
> Thank you for providing constructive comments and critiques. We believe this input has improved the quality of our work.
>
> ## Major:
>
> I) Thank you for the insightful comment about possible shortcomings. We will add this to the revised manuscript. The VIBI's explainer always returns a certain form of output (a $k$-hot vector assigned to each chunk) and the IB layer will always have a certain form (a masked input). If we understand your comment correctly, this can help to address such concerns to some extent because the explanations have a form that are at least recognizable to humans. However, the hand-crafted chunking strategies change the way the information is represented. We think the minor comment a) you made below is in line with this issue. We would appreciate it if you can make a further discussion about this issue.
>
> II) Please see our next comment.
>
> III) Thank you for carefully expressing the concerns about the experiment. The two evaluation strategies we used in the IMDB and MNIST analysis have pros and cons. The first strategy (i.e., asking participants on MTurk to infer the black-box output, like in the IMDB experiment) is less subjective and has less variation among individual participants. However, it is a proxy approach to evaluate how `good' the explanations are: the measurements could be considered as human-evaluated fidelity than interpretability. For using this strategy, we need to assume that a `good' explanation allows humans to better infer the black-box output, which is not always true. For example, inferring the black-box output based on the `good' off-digit explanations is harder than the same `good' on-digit explanations. Also, think about the case when the consistency of the explanations between similar inputs is a matter for evaluating the goodness of explanations. The second strategy (i.e. asking participants to directly score conciseness of the explanation, like in the MNIST experiment) is a direct way to evaluate how `good' the explanations are, which can evaluate the `goodness' of interpretation based on a whole aspect of goodness to HUMAN, not just based on its fidelity. But it requires qualified survey participants who can evaluate the explanations based on whether they could gain a `good' understanding of how the black-box makes decisions. We strongly believe that the participants can evaluate based on whether they could gain more understanding of how the black-box makes decisions. We first would like to note that most of them have taken three or more graduate-level ML/DL classes and have two or more papers published in ML/DL or related fields. (Please understand that we are afraid to reveal many details about them, which can be a clue to infer the author's affiliation.) Regarding the concerns that the participants may prefer explanations that lie on digits than off digits, we informed the participants that the `good' explanation does not need to be on the digits. Whether the chunks lie on or off digits, the participants actually gave higher scores if the chunks caught all necessary key characteristics of digits. Regarding the concerns that it is unclear whether they simply prefer the chunks over others or they have actually gained more understanding, we asked the participants to use the absolute scoring (NOT the relative scoring) based on the criteria in Figure 7, which can help to reduce such concerns.
>
> We hope this helps to address your concerns about the experiments.
>
> IV) We have fixed Table 2, as suggested. Regarding the approximator fidelity of VIBI and L2X, we think it gives an insight into the benefit of using the compressiveness term (i.e., I(x, t)). Note that L2X can be viewed as a special case of VIBI that has no compressiveness term (i.e., $beta = 0$). Since the Gumble-softmax trick uses the continuously-relaxed masked $\mathbf{x}$ as an input to the approximator for learning the models, the information may go through the non-selected chunks. Our experiment reported VIBI and L2X have similar approximator fidelity, which is the prediction performance of the approximator taking continuous-relaxed masked $\mathbf{x}$ as an input. However, VIBI outperforms L2X in the rationale fidelity, which is the prediction performance of the approximator taking hard-masked $\mathbf{x}$ as an input. This tells that the compressiveness term can help to prevent information leakage through the non-selected chunks. In other words, it helps the information pass mostly through the selected chunks.
>
> V) We have stated the form of the prior and KL-term in the Supplementary Material B (colored by red).
>
> ## Minor:
>
> a) Please see our response to the major comment I.
>
> b - c) Thank you for the constructive suggestions for our future work. We think the suggestions are very helpful. We also plan to extend this work by using better IB models.

---

> > ### Author Response · Authors · 2019-11-09
> > **(Continued) Thank you for providing constructive comments and critiques.**
> >
> > II) We will further explore how much does our method depends on the approximator. We will then add a thorough discussion about this issue. Both intuitively and theoretically, different models of the approximator indeed affect the quality as well as evaluated fidelity of the explanation.  We somehow observed this in the experimental analysis.  For both IMDB and MNIST, we investigated how different approximators affect the fidelity.
> >
> > For IMDB, we compared a CNN (reported in our first submission) and several approximators with recurrent nodes: RNN, GRU, LSTM, bi-directional LSTM, LSTM with a smaller number of hidden units, and bi-directional LSTM with a smaller number of hidden units. For details about the approximator structures, please see the code attached (VIBI-code-ICLR/imdb/explainer.py): CNN (line 203-221), RNN (line 135-148), GRU (line 187-201), LSTM (line 150-163), and bi-directional LSTM (line 150-163), LSTM with a smaller number of hidden units (line 176-185), and bi-directional LSTM with a smaller number of hidden units (165-174).
> >
> > As seen in Table 1, the LSTM approximator has lower fidelity than the CNN approximator in general, especially when we use a word for the chunk size. Other recurrent approximators achieved similar fidelity to the LSTM approximator.
> > ----------------------------------------------------------------
> > Table 1.
> > ----------------------------------------------------------------
> >           | Approximator        | Rationale
> >           | Fidelity                    | Fidelity
> > ----------------------------------------------------------------
> > chunk | k | CNN  | LSTM  | CNN  | LSTM  |
> > size     |    |            |            |           |             |
> > ----------------------------------------------------------------
> > sent  | 1   | 87.7   | 87.8    | 73.1   | 71.7    |
> >           |      |(0.6)   | (1.0)    | (0.8)  | (2.4)   |
> > ----------------------------------------------------------------
> > word  | 5 | 74.4   | 61.3    | 65.7   |  57.1   |
> >            |     |(0.8)   | (3.0)   | (0.8)   | (2.4)   |
> > ----------------------------------------------------------------
> > *We performed two runs for LSTM and reported the best results.
> > ----------------------------------------------------------------
> >
> > For MNIST, we compared an MLP approximator with two hidden layers (NN), a shallow 2D-CNN approximator, and a deep 2D-CNN approximator (reported in our first submission).  For details about the approximator structures, please see the code attached (VIBI-code-ICLR/mnist/explainer.py): NN (line 95-103), CNN1-2 (line 75-85). In contrast to the IMDB, there were no well-marked differences between the approximators and/or between different chunk sizes.
> >
> > Although it is too early to conclude only based on current results, it seems like the fidelity is more affected by the type of approximators (e.g. CNN or LSTM) rather than the capacity of approximators. Also, it depends on the chunk sizes.
> >
> > We will further explore and hopefully show some additional results in the rebuttal period.

---

> > > ### Comment · AnonReviewer2 · 2019-11-14
> > > **Thank you for the updated manuscript**
> > >
> > > Thank you for addressing II) Just to clarify: my main point here is that the method will produce explanations for the approximator - if the approximator does not match the model in some cases, the corresponding explanation might also not match. This should be clearly stated in the paper.
> > >
> > > I appreciate that the paper actually goes one step further an looks into approximator quality thoroughly, which I think is necessary for the reasons given above.  This will not completely eliminate the problem, but it should strongly contribute to keeping the problem under control.

---

> > > > ### Author Response · Authors · 2019-11-15
> > > > **We've updated the human evaluated interpretability.**
> > > >
> > > > Thank you for the constructive comment. We've updated the human evaluated interpretability. We've also stated the issues and discussed on the supplementary material "E3 Extra experiment - choice of approximator" as follows:
> > > >
> > > > Different approximators affect fidelity and interpretability of the explanation, which makes sense both intuitively and theoretically. In theory, explanation is obtained via solving the problem in Equation (2) where $I ( \mathbf{t}, \mathbf{y} )$ represents the approximator. Different models of the approximator result in different $I ( \mathbf{t}, \mathbf{y} )$ and optimizing the information bottleneck objective amounts to learning a decent approximator. Intuitively, different models of the approximator indeed affect the quality as well as evaluated fidelity of the explanation. If the model of approximator has low capacity, $I ( \mathbf{t}, \mathbf{y} )$ and fidelity tend to be low. To remedy this, explainer has to generate less brief explanations; vice versa.
> > > >
> > > > In addition to the approximator capacity, the chunk sizes, more precisely chunk types, can be an essential factor for choosing approximators. In the IMDB experiment, we investigated how the two approximators affect the fidelity and interpretability. As seen in Table 9, the LSTM approximator has lower fidelity than the CNN approximator when we use a word for the chunk size, while both approximators achieve similar fidelity when we use a sentence for the chunk size. When we use a sentence as a cognitive chunk, the LSTM achieves better interpretability than CNN, which is more substantial than when we use a word as a cognitive chunk. This may indicate that a sentence (as an explanation) should be processed with an approximator that resembles the black-box model, while a group of single words is relatively free from such a constraint. This may be because a sentence is more similar to the original input (i.e., movie review -- one or two paragraphs) than a group of single words.
> > > > ----------------------------------------------------------------
> > > > Table 9.
> > > > --------------------------------------------------------------------------------------------
> > > >           | Approximator        | Rationale          | Interpretability
> > > >           | Fidelity                    | Fidelity               |
> > > > --------------------------------------------------------------------------------------------
> > > > chunk | k | CNN  | LSTM  | CNN  | LSTM  | | CNN  | LSTM  |
> > > > size     |    |            |            |           |             |  |           |             |
> > > > --------------------------------------------------------------------------------------------
> > > > sent  | 1   | 87.7   | 87.8    | 73.1   | 71.7    |    45.1   |   65.9   |
> > > >           |      |(0.6)   | (1.0)    | (0.8)  | (2.4)   |
> > > > --------------------------------------------------------------------------------------------
> > > > word  | 5 | 74.4   | 61.3    | 65.7   |  57.1   |     49.4     |    51.3   |
> > > >            |     |(0.8)   | (3.0)   | (0.8)   | (2.4)   |                 |
> > > > --------------------------------------------------------------------------------------------
> > > > *We performed two runs for LSTM and reported the best results.
> > > > ----------------------------------------------------------------
> > > >
> > > > In practice, we recommend choosing the approximator that matches the black-box model, especially if a cognitive chunk resembles the original input. We then recommend starting with an approximator with enough capacity and reducing it as far as the approximator achieves a target fidelity because a large capacity usually makes it hard to learn.

---

> > ### Comment · AnonReviewer2 · 2019-11-14
> > **Thank you for the thorough response and updated manuscript**
> >
> > With the improvements, rebuttal and clarifications I am slightly leaning towards acceptance of the paper. I still think the MNIST experiments could be improved, which is why I am hesitant to give a high score.
> >
> > I) Just to clarify: my main point here was that a 'good' explanation is not necessarily the most compressed one (though there is certainly a relation between the two) and this should be clearly stated in the paper. Hand-crafting cognitive chunks helps with making sure that the explanations are easily understandable, but do not (necessarily) allow for maximum compression anymore. I think overall the handcrafted cognitive chunks are interesting - all I wanted to point out was that they lead to a deviation from the strict theory, which means that some appealing theoretical properties might not necessarily hold in practice.
> >
> > II) see comment below
> >
> > III) I still don't fully agree. Just because participants have a degree in CS (or taken some lectures) does not necessarily mean that they are able to objectively identify whether a shown set of cognitive chunks is a better or worse explanation. A good explanation allows the judge to make some non-trivial *predictions* of how the system works (as you phrase it 'evaluate how good the explanations are'). Ideally experiments are set up in a way to test such predictions (e.g. the way this was performed in the IMDB experiment) and/or allow the judges to interact with the system. I don't think the MNIST setup is completely wrong - it's just the weakest part of the experimental section. Saliency maps can easily be misleading and mis-interpreted, even by ML researchers, if they are not used for thorough hypothesis testing (see e.g. https://openreview.net/forum?id=rkl3m1BFDB). While I'm not saying that this is the case in the MNIST experiment, there is currently no way to rule this out, which is why I'm still a bit sceptical.
> >
> > IV and V) Thanks for the improvements and clarification.

---

> > > ### Author Response · Authors · 2019-11-15
> > > **Thank you for the thoughtful responses.**
> > >
> > > Thank you for the insightful and constructive comments.
> > >
> > > I-1) We've discussed this issue and potential shortcomings to the revised manuscript as follows:
> > >
> > > The information bottleneck framework provides a theoretical background that the bottleneck captures a minimal sufficient statistic, i.e. the most compressed representation that captures all the possible (i.e. sufficient) amount of information about output. For finite $\beta$, the bottleneck approximates such a minimal sufficient statistic.
> > >
> > > However, the way this information is represented may have a substantial effect on interpretability. VIBI helps to address this issue to some extent by always returning a certain form of output (i.e., a $k$-hot vector $z$ assigned to each chunk) and having a certain form of the information bottleneck layer (i.e., a masked input) so that it makes sure that the explanations are easily understandable to humans. In practice, such a chunking strategy leads to a deviation from the strict theory that a 'good' explanation is the most compressed one but helps to achieve better interpretability in practice.
> > >
> > >
> > > I-2) Regarding the hand-crafted chunking strategies (minor comment (a)), we've added several qualitative examples with different chunk sizes to illustrate how the method depends on the hand-crafting chunking strategy (Figure 9 in the supplementary material). We have also added some discussion on how to decide the chunk size and how hard/easy this might be for different domains as follows:
> > >
> > > The chunk size changes the way the information is represented in the information bottleneck, which can affect interpretability. For choosing the chunk size, a greater fidelity does not always mean better interpretability. For example, in the MNIST experiment, the models with the chunk size $1 \times 1$ have achieved better fidelity than those with $2 \times 2$ and $4 \times 4$ under the same number of pixels that are used for the information bottleneck, but $1 \times 1$ is less recognizable than $2 \times 2$ or $4 \times 4$ (Figure~9).

---

> > > > ### Author Response · Authors · 2019-11-15
> > > > **(Continued) Thank you for the thoughtful responses.**
> > > >
> > > >
> > > > III) We sincerely thank you for making thoughtful discussions. We would like to first clarify responses we made before and then discuss further.
> > > >
> > > >  - We simply wanted to point out that the participants were understanding that the task was to evaluate whether the explanations are good for getting an insight into how the black-box model works. (Although we agree that this does not completely rule out the possibility that they make a misinterpretation or misleading.)
> > > >
> > > > - We also would like to point out followings:
> > > > Only a few works in interpretable ML (IML) have evaluated interpretability by asking humans. Of course, there are several concerns that such evaluations are not reproducible so that it cannot be a good way to evaluate interpretability. However, with a rigorous experimental design and qualified human participants, such evaluations are a valuable source to assess the quality of explanation that the computational metric based approach couldn't provide. Especially, since IML aims to provide an explanation about how black-box works to HUMANS, human-based evaluation is inevitable. In this aspect, we believe the work done in here is valuable as well as our discussion regarding these issues.
> > > >
> > > > - Deciding which strategy provides a better explanation depends on how we define the `goodness' of interpretation. But, there is no consensus regarding this in the community yet. We think this is essentially about what is a `good' explanation (and hence what is the right way--or better way--to evaluate the interpretability).

---

### Official Review · AnonReviewer1 · 2019-10-24
**Official Blind Review #1**

**Rating:** 6

**Review:**

The paper proposes a method to learn an explanation of black-box systems from its outputs. The method is based on the information bottleneck as the objective function is designed to measure mutual information between input x, system output y, and narrowed information of input t. t is constructed by filtering x with maintaining interpretability of y, so that it is finally assumed as the explanation of the system extracted by the proposed method.

The paper is well motivated and well written. Enough experiments were conducted to assess the advantage of the proposed method in the classification tasks. It looks a good paper.

Maybe the paper is focused on only tasks that the predictor does not generate much information, such as classification. It is still unclear how the proposed method work when it is applied to the output-rich models, i.e., the model should keep as much information as inputs.

The proposed method automatically selects some important chunks from inputs, but the chunks still rely on some task-specific hand-crafted chunking strategies. The paper also conducted some experiments by changing the strategy, but it is still unclear what is the important criteria.

It is also good to show how actually the thickness of the bottleneck (controlled by k) works in actual cases, e.g., showing results for the same example with moving k.

Trivial comments:
* The example in 3.2 "great, great" and "great, thought provoking" looks still ambiguous to explain what the section want to say.
* "x_i \times z_j" in p.iv looks ambiguous.
* z_j^* in 3rd eqn. of p.v should take l: z_j^{*(l)}
* the max operator over l in 3rd eqn. of p.v looks to hide other values than the highest one (specifically, the L1 norm of z_j^* does not become k by this eqn. as the k-hot vector does). Summation looks intuitively better than max. Could you explain how this eqn. was constructed?
* f(.) in the 4th eqn. of p.v may be undefined in the main text.
* \beta_1 of Adam looks to be set to not a standard value (0.9). Is there any reason?

**Experience Assessment:**

I do not know much about this area.

**Review Assessment: Checking Correctness Of Derivations And Theory:**

I assessed the sensibility of the derivations and theory.

**Review Assessment: Checking Correctness Of Experiments:**

I assessed the sensibility of the experiments.

**Review Assessment: Thoroughness In Paper Reading:**

I read the paper at least twice and used my best judgement in assessing the paper.

---

> ### Author Response · Authors · 2019-11-13
> **Thank you for the detailed comments, and we hope that our revisions address your concerns.**
>
> Thank you for the detailed comments, and we hope that our revisions address your concerns.
>
> Comment1: Maybe the paper is focused on only tasks that the predictor does not generate much information, such as classification. It is still unclear how the proposed method work when it is applied to the output-rich models, i.e., the model should keep as much information as inputs.
>
> Answer 1 : (Note - It is hard for us to understand what it means by 'the output-rich models.' If you can give some examples of the output-rich models, it will be helpful.) If we understand your comment correctly, your concern is that (due to the compression procedure) our model may not able to pass enough information to the outputs in the output-rich models. But this is not the case. The IB framework (Tishby 2015, 2000) provides a theoretical background that IB captures a minimal sufficient statistic, i.e. the most compressed representation that captures all the possible (i.e. sufficient) amount of information about output. Our IB approximates such a minimal sufficient statistic, so it approximately covers a sufficient amount of information about (both rich and non-rich) outputs.
>
> Comment 2: The proposed method automatically selects some important chunks from inputs, but the chunks still rely on some task-specific hand-crafted chunking strategies. The paper also conducted some experiments by changing the strategy, but it is still unclear what is the important criteria.
>
> Answer 2: Please see our next comment.
>
> Comment 3: It is also good to show how actually the thickness of the bottleneck (controlled by k) works in actual cases, e.g., showing results for the same example with moving k.
>
> Answer 3: Thank you for the suggestion. We have added several qualitative examples with moving $k = 4, 6, 10, 20$ to the revised manuscript (Figure 8 in the supplementary material). Obviously, the compressiveness of explanations depends on the sparsity $k$ (i.e., the number of cognitive chunks to be selected). A larger $k$ allows the information bottleneck to convey more information about output, but it gives less compressive explanations than a smaller $k$. For deciding $k$, we recommend choosing the minimum possible $k$ that achieves a target fidelity because an unnecessarily large $k$ can make redundant explanations (i.e., chunks). Figure 8 shows how our method works under different sparsity. When we increase $k$, VIBI tends to select chunks that are the same or nearby the previously selected chunks and additionally select patches that catch new characteristics of digits. The most important explanations tend to be selected again at a larger $k$.
>
> We will add the above discussion on how our method works under different sparsity and the choice of $k$ to the revised manuscript. If you have any, we are happy to have a further discussion about this.
>
> Trivial comments:
>
> 1) There were typographical errors. We've revised "great, thought provoking" to "great, thought-provoking". Also, we've revised the whole paragraph as follows:
>
> For example, consider a movie review where "great" occurs a lot and two explanations in judging the sentiment of the review: "great, great" and "great, thought-provoking". They have the same level of sparsity ($k = 2$), but the former has semantic redundancy. In this case, MI helps to choose a better explanation. The first explanation has a larger MI with the input document. The second explanation has smaller MI and hence is more brief and preferable.
>
> 2) Intuitively, $\mathbf{t}$ is the masked input of $\mathbf{x}$ having the masking indicator $\mathbf{z}$. We hope Figure 1B help you to understand this formula. For example, the j-th ($j = 2$) word "cough" is represented by three features $[x_4, x_5, x_6]$ and selected by the indicator $z_2 = 1$. Now, its masked input is $[t_4, t_5, t_6] = [x_4 \times z_2, x_5 \times z_2, x_6 \times z_2] = [x_4, x_5, x_6]$. Similarly, the j-th ($j = 3$) word "a" is represented by $[x_7, x_8, x_9]$ but not selected (i.e. $z_3 = 0$). Now, its masked input is $[t_7, t_8, t_9] = [x_7 \times z_3, x_8 \times z_3, x_9 \times z_3] = [0, 0, 0]$.
>
> 3) $z_j^{*}$ should NOT take $l$. The $l$ should be removed, since $z_j^{*}$ is the maximum value $c_j^{(l)}$ over all $l$'s.
>
> 4) Note that $\mathbf{z}^{*}$ is a continuous-version of the k-hot vector $\mathbf{z}$. For example, we can have $\mathbf{z}^{*} = [0.99, 0.01, 0.08, 0.90]$ as a continuous-version of the k-hot ($k = 2$) vector $\mathbf{z} = [1, 0, 0, 1]$. Here, $z_1^{*} = 0.99, z_2^{*} = 0.01, z_3^{*} = 0.08, z_4^{*} = 0.90$ and the L1 norm of $\mathbf{z}^{*}$ (NOT $\mathbf{z}_j^{*}$) will be approximately $k (= 2)$ which is the L1 norm of $\mathbf{z}$. The L1 norm of $\mathbf{z}_j^{*}$ should be approximately 1, hence "max" is the right choice.
>
> 5) Thank you for pointing out. We've added the definition of $f(.)$. Also, we've revised the whole formula to make it more clear.
>
> 6) $\beta_1$ of Adam is the default value provided by PyTorch.

---

> > ### Author Response · Authors · 2019-11-14
> > **(Continued) Thank you for the detailed comments, and we hope that our revisions address your concerns.**
> >
> > Comment 2: The proposed method automatically selects some important chunks from inputs, but the chunks still rely on some task-specific hand-crafted chunking strategies. The paper also conducted some experiments by changing the strategy, but it is still unclear what is the important criteria.
> >
> > Answer 2: We have added several qualitative examples with different chunk sizes to illustrate how the method depends on the hand-crafting chunking strategy (Figure 9 in the supplementary material). We have also added some discussion on how to decide the chunk size and how hard/easy this might be for different domains as follows:
> >
> > The chunk size changes the way the information is represented in the information bottleneck, which can affect interpretability. For choosing the chunk size, a greater fidelity does not always mean better interpretability. For example, in the MNIST experiment, the models with the chunk size $1 \times 1$ have achieved better fidelity than those with $2 \times 2$ and $4 \times 4$ under the same number of pixels that are used for the information bottleneck, but $1 \times 1$ is less recognizable than $2 \times 2$ or $4 \times 4$ (Figure 9).
> >
> > Choosing the chunk size is not trivial, and the strategy varies for different domains (except the case that each (raw) feature itself is cognitive (e.g., age, gender) so that the chunk size should be 1). Obviously, the chunk size should be larger than the size of the smallest cognitive unit. For example, in the IMDB experiment, the smallest cognitive unit was set to 50 because 50 raw features compose a word. In our sentiment prediction task, using a word as the chunk size can be enough to explain the decision (i.e., positive/negative sentiment); to infer the sentiment, we do not need to know a whole sentence. However, we may need to use a phrase or sentence as the chunk size for complex tasks such as disease diagnosis or patient need detection since decisions on such tasks are made based on contextual information. In the MNIST experiment, we set the chunk size to be $4 \times 4$ because the digits' stroke width is about to $4$. However, we may need to use different chunk sizes, for example, for automated visual inspection for identifying defects on a factory production line. In this case, the chunk size can be set to the average defect size on the production line.

---

### Official Review · AnonReviewer3 · 2019-10-25
**Official Blind Review #3**

**Rating:** 3

**Review:**

"an information theoretic principle, information bottleneck principle" in the abstract is quite redundant with the use of 'principle' twice

'"great, great" and "great, thought provoking". They have the same level of sparsity.' What kind of sparsity are you referring to with this example? Why can't sparsity reduce semantic redundancy? Please explain further.

"However, the first explanation has a large MI with the input document where "great" occurs a lot." What example input document are you referring to?

You should save the explanation of how your method in Equation 2 differs from the original information bottleneck of Equation 1 until after you have actually written out Equation 2. As it is now, you are referencing Equation 2 before it has been seen.

I find Equation 2 confusing. Is it possible to make the dependence of the expression on z more explicit. It isn't clear from the equation itself how p(z|x) influences either quantity in Equation 2. Perhaps you should wait to introduce this equation until you have first explained how z relates to x and t. As it is, z is not clearly defined. I can gather information about it from the figure, from how you describe the difference in your method from the original information bottleneck, but the relationship is not clear enough by reading only the text of the paper before Equation 2 is presented.

Can you explain briefly how your "hierarchical LSTM" works in the main text of the paper? Its an unusual enough term that I would want to see a citation or brief explanation right away rather than having it deferred to the Appendix. Why not use a state-of-the-art model for IMDb? Are you not using the standard splits for IMDb? the

In Appendix B.1 "output vector is averaged and followed by log-softmax calculation. The final layer is formed to return a log-probability indicating which cognitive chunks should be taken as an input to the approximator" The single output vector of the biLSTM is averaged and followed by log-softmax? the final layer is formed? What does this mean?

I find the phrasing of "Negative Sentiment if any negative words" and the corresponding title for positive in Fig 2 confusing. What do you mean by "if any"? The phrasing makes it sound like the prediction of the model somehow depends on a logical step based on whether there are any negative/positive words found.

I find the lack of a comparison to some kind of attentional method somewhat glaring in the IMDb example, since I would expect that many classifiers with attention would simply attend to the same words. What does your method give us that attention would not?

The same can be said for the MNIST example regarding an attention map.

"by the human intelligences" sounds quite robotic

Can you provide some sense of inter annotator agreement for labeling the images and sentences?

It does seem that there is key information like the definition of approximator fidelity in the appendices which is crucial to actually understanding the paper.

**Experience Assessment:**

I have read many papers in this area.

**Review Assessment: Checking Correctness Of Derivations And Theory:**

I assessed the sensibility of the derivations and theory.

**Review Assessment: Checking Correctness Of Experiments:**

I carefully checked the experiments.

**Review Assessment: Thoroughness In Paper Reading:**

I read the paper at least twice and used my best judgement in assessing the paper.

---

> ### Author Response · Authors · 2019-11-15
> **Thank you for the detailed comments, and we hope that our revisions address your concerns.**
>
> Thank you for the detailed comments, and we hope that our revisions address your concerns. Here, we first address your 1 - 5th comments. We will address the 6 - 11th in the next post.
>
> Comment 1 and 2) '"great, great" and "great, thought provoking". They have the same level of sparsity.' What kind of sparsity are you referring to with this example? Why can't sparsity reduce semantic redundancy? Please explain further. "However, the first explanation has a large MI with the input document where "great" occurs a lot." What example input document are you referring to?
>
> Answer 1 and 2) There were typographical errors. We've revised "great, thought provoking" to "great, thought-provoking". (Note that "thought-provoking" is a word not two words (https://www.merriam-webster.com/dictionary/thought-provoking)). Here, the sparsity indicates the number of words ($k = 2$). For better understanding, we have revised the section as follow:
>
> For example, consider a movie review where "great" occurs a lot and two explanations in judging the sentiment of the review: "great, great" and "great, thought-provoking". They have the same level of sparsity ($k = 2$), but the former has semantic redundancy. In this case, MI helps to choose a better explanation. The first explanation has a larger MI with the input document. The second explanation has smaller MI and hence is more brief and preferable.
>
> Comment 3) You should save the explanation...
>
> Answer 3) Thank for pointing out. We have moved the explanation to after Equation 2.
>
> Comment 4) I find Equation 2 confusing. Is it possible to make the dependence of the expression on z more explicit. It isn't clear from the equation itself how p(z|x) influences either quantity in Equation 2. Perhaps you should wait to introduce this equation until you have first explained how z relates to x and t. As it is, z is not clearly defined. I can gather information about it from the figure, from how you describe the difference in your method from the original information bottleneck, but the relationship is not clear enough by reading only the text of the paper before Equation 2 is presented.
>
> Answer 4) Thank you point out. In the revised manuscript, we have added the definition of $z$ and moved the paragraph that explains $z$ and its relationship $x$ and $t$ right after the equation (2)'s paragraph.
>
> Comment 5) Can you explain briefly how your "hierarchical LSTM" works in the main text of the paper? Its an unusual enough term that I would want to see a citation or brief explanation right away rather than having it deferred to the Appendix. Why not use a state-of-the-art model for IMDb? Are you not using the standard splits for IMDb? the
>
> Answer 5) (1) We have added a sentence to it. We did not include much information about the black-box models because black-box model itself is NOT essential and how good the black-box model is NOT of interest. This is the reason that we do not use the state-of-the-art model for IMDB. What we are interested in this paper is to EXPLAIN black-box models no matter how good the black-box models are. For example, in our TCR experiment, we explained a black-box model having a BAD performance and showed that VIBI's explanation could be used to improve the performance. In the IMDB and MNIST experiment, we used CNN for MNIST and LSTM for IMDB because they are well-known so that people already have some idea how the models work, which can help to assess the quality of explanations provided by VIBI and others themselves. (2) It is not clear what "splits" you are asking in the last sentence. We are guessing you are talking about dataset splits for training/validation/test sets. (Please clarify if this is not what you intended.) The original split provided by PyTorch is 25,000 for training and 25,000 for testing, which does not have samples for validation. Therefore, we split the 25,000 test set into 12,500 validation and 12,500 test sets. Similarly, we also did it for the MNIST.

---

> > ### Author Response · Authors · 2019-11-15
> > **(Continued) Thank you for the detailed comments, and we hope that our revisions address your concerns.**
> >
> > Comment 6) In Appendix B.1 "output vector is averaged and followed by log-softmax calculation. The final layer is formed to return a log-probability indicating which cognitive chunks should be taken as an input to the approximator" The single output vector of the biLSTM is averaged and followed by log-softmax? the final layer is formed? What does this mean?
> >
> > Answer 6) We have re-wrote the paragraph for a better description as follows. Also, we would like to refer to the code attached for detailed information about the structures.
> >
> > For the explainer, we use a bidirectional layer that returns multiple output vectors, each of which corresponds to a recurrent unit. Each element in the output vectors are averaged over all units, and then the averaged output vector is followed by log-softmax calculation. As a result, the explainer returns a vector of log-probabilities, each of which indicates whether or not each cognitive chunk will be selected as an input to the approximator.
> >
> > Comment 7) I find the phrasing of "Negative Sentiment if any negative words" and the corresponding title for positive in Fig 2 confusing. What do you mean by "if any"? The phrasing makes it sound like the prediction of the model somehow depends on a logical step based on whether there are any negative/positive words found.
> >
> > Answer 7) We have updated Figure 2. The phrases describe the insight we could get from VIBI's explanation, which intuitively makes sense. For example, from VIBI's explanation, we could infer that the black-box model predicted the movie review to have negative sentiment because several negative words exist in the review. Hence what you saying "the prediction of the model somehow depends on a logical step based on whether there are any negative/positive words found" is what we think VIBI says to us.
> >
> > Comment 8) I find the lack of a comparison to some kind of attentional method somewhat glaring in the IMDb example, since I would expect that many classifiers with attention would simply attend to the same words. What does your method give us that attention would not? The same can be said for the MNIST example regarding an attention map.
> >
> > Answer 8) VIBI is system agnostic, while attention methods are system-specific. The system-agnostic approaches have a greater advantage over system-specific approaches because they explain any black-box systems (even for traditional ML systems such as random forest and for human-performing decision systems). See [1], [2] for other desirable aspects of the agnostic approaches. Furthermore, VIBI aims to find a brief but comprehensive explanation by both (1) minimizing the mutual information between $x$ and $t$ and (2) controlling the sparsity level $k$. The attention-based approaches do none of them. They only aim to minimize a loss function, which only helps to optimize informativeness of the explanations. Without consideration of compressiveness, we would expect that the approaches would find less concise, more redundant explanations.
> >
> > [1] Ribeiro et al. Model-agnostic interpretability of machine learning, ICML Workshop on Human Interpretability in Machine Learning (2016)
> > [2] Ribeiro et al. Why should i trust you?: Explaining thepredictions of any classifier, ACM SIGKDD (2016)
> >
> > Comment 9) "by the human intelligences" sounds quite robotic
> >
> > We've changed it to "... by the humans ..."
> >
> > Comment 10) Can you provide some sense of inter annotator agreement for labeling the images and sentences?
> >
> > We first note that IMDB and MNIST are publicly available datasets, which are not collected by us. Hence it is hard to know the details about their labeling strategy. As far as we know, the IMDB label is NOT annotated by annotators; the movie reviews are labeled based on the scores that are given by the persons who actually wrote the review. See the followings: "In the labeled train/test sets, a negative review has a score <= 4 out of 10, and a positive review has a score >= 7 out of 10. Thus reviews with more neutral ratings are not included in the train/test sets." (ref - https://www.kaggle.com/iarunava/imdb-movie-reviews-dataset). The MNIST label seems NOT to be annotated by annotators. The hand-digits are collected by re-mixing the samples collected through a handwriting sample form. The form gives specific numbers and asks humans to write down the numbers. Hence, no need to annotate the labels. (ref - https://www.nist.gov/sites/default/files/documents/srd/nistsd19.pdf)
> >
> > Comment 11) It does seem that there is key information like the definition of approximator fidelity in the appendices which is crucial to actually understanding the paper.
> >
> > We'd say it is NOT. The definition of approximator fidelity and rationale fidelity of VIBI is in the main text (first paragraph, page ix) and the appendices. The details about how the fidelity of other baseline methods is evaluated are in the appendices only.

---

### Author Response · Authors · 2019-11-15
**Major Updates**

We thank all reviewers for their thoughtful comments which helped us to improve the manuscript. All major and minor changes are colored with red in the revised manuscript. The major revisions we made are as follows:

1) We've added extra experiments and made discussions on
 - the choice of $k$
 - the choice of the chunk size
 - the choice of the approximator
which can be found in the supplementary material "E. Extra Experiments - 1 to 3."

2) We've added a discussion in the "Conclusion" section about a gap between the theory and practice for finding a `good' explanation.

3) There is an on-going discussion with Reviewer2 essentially about what is a `good' explanation and hence what is the right way---or better way---to evaluate interpretability by humans.

---

### Decision · Program_Chairs · 2019-12-19

**Decision:**

Reject

**Comment:**

The authors present a system-agnostic interpretable method based on the idea of that provides a brief (=compressed) but comprehensive (=informative) explanation. Their system is build upon the idea of VIB. The authors compare against 3 state-of-the-art interpretable machine learning methods and the evaluation is terms of interpretability (=human understandable) and fidelity (=accuracy of approximating black-box model). Overall, all reviewers agreed that the topic of model interpretability is an important one and the novel connection between IB and interpretable data-summaries is a very natural one.

This manuscript has generated a lot of discussion among the reviewers during the rebuttal and there are a number of concerns that are currently preventing me from recommending this paper for acceptance. The first concern relates to the lack of comparison against attention methods (I agree with the authors that this is a model-specific solution whereas they propose a model-agnostic one), however attention is currently the elephant in room and the first thing someone thinks of when thinking of interpretability. As such, the authors should have presented such a comparison. The second concern relates to the human evaluation protocol which could be significantly improved  (Why 100 samples from all models but 200 for VIBI? Given the small set of results, are these model differences significant? Similarly, assuming that we have multiple annotations per sample, what is the variance in the annotations?).

This paper is currently borderline and given reviewers' concerns and the limited space in the conference program I cannot recommend acceptance of this paper.